# Exactly Solvable Floquet Dynamics for Conformal Field Theories in Dimensions Greater than Two

Diptarka Das[1], Sumit R. Das[2], Arnab Kundu[3,4], Krishnendu Sengupta[5],

**1** Department of Physics, Indian Institute of Technology, Kanpur, UP 208016, India.
**2** Department of Physics and Astronomy, University of Kentucky, Lexington, KY 40506, U.S.A.
**3** Saha Institute of Nuclear Physics, 1/AF Bidhannagar, Kolkata 700064, India.
**4** Homi Bhaba National Institute, Training School Complex, Anushaktinagar, Mumbai 400094, India.
**5** School of Physical Sciences, Indian Association for the Cultivation of Science, Jadavpur, Kolkata 700032, India.

## Abstract

**We find classes of driven conformal field theories (CFT) in $d+1$ dimensions with $d > 1$, whose quench and floquet dynamics can be computed exactly. The setup is suitable for studying periodic drives, consisting of square pulse protocols for which Hamiltonian evolution takes place with different deformations of the original CFT Hamiltonian in successive time intervals. These deformations are realized by specific combinations of conformal generators with a deformation parameter $\beta$; the $\beta < 1$ ($\beta > 1$) Hamiltonians can be unitarily related to the standard (Lüscher-Mack) CFT Hamiltonians. The resulting time evolution can be then calculated by performing appropriate conformal transformations. For $d \leq 3$ we show that the transformations can be easily obtained in a quaternion formalism; we use this formalism to obtain exact expressions for the fidelity, unequal-time correlator, and the energy density for the driven system for $d = 3$. Our results for a single square pulse drive cycle reveal qualitatively different behaviors depending on the value of $\beta$, with exponential decays characteristic of heating for $\beta > 1$, oscillations for $\beta < 1$ and power law decays for $\beta = 1$. When the Hamiltonians in one cycle involve generators of a single $SL(2, R)$ subalgebra we find fixed points or fixed surfaces of the corresponding transformations. Successive cycles lead to either convergence to one of the fixed points, or oscillations, depending on the conjugacy class. This indicates that the system can be in different dynamical phases as we vary the parameters of the drive protocol. Our methods can be generalized to other discrete and continuous protocols. We also point out that our results are expected to hold for a broader class of QFTs that possesses an $\mathbf{SL}(2, C)$ symmetry with fields that transform as quasi-primaries under this. As an example, we briefly comment on celestial CFTs in this context.**

# 1 Introduction

The study of time evolution of a driven quantum system with a time dependent Hamiltonian is a valuable tool for gaining insight into its non-equlibrium properties. Useful drive protocols include quantum quench, ramp, and periodic or quasi-periodic drive protocols [1–4]. Recently, properties of quantum systems driven via periodic protocols have been most intensely studied; such systems undergo evolution governed by Hamiltonian periodic in time with a time period $T$. Interestingly, at strobosocopic times $t = nT$, where $n$ is an integer, the evolution operators of such systems can be written as $U(nT, 0) = \exp[-iH_F nT]$; the corresponding dynamics is then completely controlled by the Floquet Hamiltonian $H_F$.

However, most of these studies require numerical work: analytically tractable models are limited to free field theories or two dimensional conformal field theories. For the latter class of theories, the underlying infinite dimensional symmetry algebra provides a powerful tool to calculate physical quantities of interest. One example involves a sudden quench from a massive theory to a two dimensional CFT whose IR properties can be well

approximated by a Cardy-Calabrese state [5], and the time evolution can be calculated by using conformal mapping. A more recent example involves quench and Floquet dynamics in a two dimensional CFT [6–17]: once again the time evolution can be obtained by conformal maps. No such analytical results are known in higher dimensions, and one has to typically resort to numerical calculations which are often limited by finite size effects [1].

In this paper we demonstrate that a class of time dependent problems can be exactly studied in conformal field theories in *arbitrary number of dimensions* and lead to interesting non-trivial dynamics. Our work is inspired by the work of [6] who studied a $1 + 1$ dimensional CFT on a strip which starts from the usual CFT Hamiltonian and is quenched by a sine squared deformed (SSD) one. The SSD-CFT$_2$ has been studied previously in [18–21]. Both these Hamiltonians belong to the global part of the Virasoro algebra. The time evolution then becomes a Möbius transformation. The global part of the Virasoro algebra is (in Euclidean signature) $SO(3,1)$ (generated by $L_0, L_{\pm 1}$ and their hermitian conjugates). Floquet generalizations to periodically driven CFTs by a sequence of the usual CFT Hamiltonian and SSD Hamiltonian for successive time durations $T_i$ were studied in [12, 14, 17]. Generalizations of the SSD to other $SL_2$ subgroups of Virasoro were studied in [15]. Holographic interpretations were investigated in [7–9,11] and from a different perspective in [22].

In higher dimensions $(d+1)$, the conformal algebra is $SO(d+2,1)$. This suggests that a natural generalization to higher dimensional systems can be obtained by considering different generators of the conformal algebra as Hamiltonians and considering sequences of non-commuting Hamiltonians in successive time intervals. Similar to the two dimensional case, the dynamics is expected to be equivalent to conformal transformations.

In this work we consider conformal field theories on $S^d \times$ time. The class of Hamiltonians are combinations of generators which belong to $SL(2,R) \sim SU(1,1)$ subrgoups of the conformal group. More specifically we consider

$$H(\beta, \Pi) = 2iD + i\beta(K_\mu + P_\mu)\Pi^\mu , \tag{1}$$

where the operators $D, K_\mu, P_\mu$ $(\mu, \nu = 0, \cdots d)$ are generators of conformal transformations and $\Pi^\mu$ is a projector along $X^\mu$-direction. Note that there are $(d+1)$ such different $\mathrm{SL}(2,R)$ subalgebras corresponding to $(d+1)$-inequivalent choices of $\Pi^\mu$. This provides a wider class of drive protocols where one can use different members of the class of Hamiltonians $H(\beta, \Pi)$, for different time intervals.

In terms of the energy momentum tensor, the Hamiltonian may be written as

$$H(\beta, \Pi) = 2\int d\Omega_{d-1} \, [1 + \frac{1}{2}\beta Y_\mu \Pi^\mu] T_{ww} \tag{2}$$

where $Y^\mu$ denote coordinates on a $R^{d+1}$, where the sphere $S^d$ is the surface $Y^\mu Y_\mu = 1$, $d\Omega_{d-1}$ denotes the volume element on $S^d$ and $w$ denotes the Euclidean time.

As will be shown below, when $\beta < 1$ a Hamiltonian of the form (1) can be transformed to the standard CFT Hamiltonian $(2iD)$ (on the plane), while for $\beta > 1$ it can be transformed to the Lüscher-Mack type Hamiltonian $P_\mu + K_\mu$.

As is standard in CFT's, it is convenient to first work in Euclidean signature and perform a Weyl transformation to a plane $R^{d+1}$

$$ds^2 = dw^2 + d\Omega_d^2 = \frac{1}{r^2} \left[ \delta_{\mu\nu} dx^\mu dx^\nu \right] \tag{3}$$

---

[1]A different class of problems involving fast smooth quenches in conformal field theories in arbitrary dimension can be addressed using conformal properties [23, 24].

where $x^\mu$ are cartesian coordinates on $R^{d+1}$ and

$$r^2 = \delta_{\mu\nu} x^\mu x^\nu = e^w. \tag{4}$$

The operator $(2iD)$ becomes the dilatation operator on the plane, while $P_\mu$ generates translations on the plane. Acting on functions on $R^{d+1}$ the conformal generators are represented by the following differential operators

$$
\begin{aligned}
D &= -ix_\mu \partial_\mu, \quad P_\mu = -i\partial_\mu, \quad K_\mu = -i(2x_\mu(x_\nu \partial_\nu) - r^2 \partial_\mu), \\
L_{\mu\nu} &= -i(x_\mu \partial_\nu - x_\nu \partial_\mu) .
\end{aligned}
\tag{5}
$$

For a fixed $\mu$ the generators $D, P_\mu, K_\mu$ form an SU(1,1) subalgebra of the Euclidean conformal algebra SO$(d+2,1)$. There are $d+1$ such subgroups corresponding to different choice of $\mu$. The SU(1,1) nature of these subgroups can be understood by noting that the commutation of these generators satisfy

$$[D, K_\mu] = -iK_\mu, \quad [D, P_\mu] = iP_\mu, \quad [K_\mu, P_\mu] = 2iD . \tag{6}$$

Our strategy for calculating the response is analogous to the earlier works in $1 + 1$ dimensions. We will perform a Weyl transformation to $R^{d+1}$, where the expressions for the generators and the resulting conformal transformation become simple, calculate the time evolution by evaluating the corresponding conformal transformation, transform back to $S^d \times$ time and finally continue to real time. While this is straightforward in principle, this becomes quickly cumbersome in practice for $d > 1$.

The key technical tool which facilitates our calculations is the fact that when $d \le 3$, a point on $R^{d+1}$ can be represented by a quaternion and the action of finite conformal transformations take a simple form, generalizing Möbius transformations on the complex plane to Möbius transformations on the field of quaternions, $SL(2, H)$. In general, the parameters of these transformations are themselves quaternions. However, when the transformation belongs to an $SL(2, R)$ subgroup of the conformal group one can judiciously choose the quaternion representation such that these parameters become real numbers. While this simplifies our calculations considerably, this necessitates switching quaternionic representations whenever we switch from one $SL(2, R)$ subgroup to another. In this work, we provide explicit results for $3 + 1$ dimensions, though it should be emphasized that the framework is completely general for any $d \le 3$.

In the study of Floquet dynamics in $d = 1$ a key step was to determine the fixed points of Möbius transformations on the complex plane and express the transformation in "normal form". In this form it is possible to determine the behavior of a point on the plane after arbitary number $n$ of Floquet cycles. The behavior for large $n$ depends on the conjugacy class of the Möbius group, with points converging to one of the fixed points for a hyperbolic class, oscillating for elliptic class and displaying maginal behavior for the parabolic class. The conjugacy class of the transformation depends on the parameters of the period drive protocol and the different behavior of points on the plane result in different dynamical phases, which are heating (for hyperbolic) and non-heating (for elliptic).

In higher dimensions $d = 2, 3$ one needs to look for fixed points or surfaces of quaternionic Möbius transformations. However, as mentioned above, if the transformation involves a definite $SL(2, R)$ subgroup, the parameters of these Möbius transformations can be chosen to be real numbers. We show that in this case there are fixed points for hyperbolic transformations and fixed surfaces for elliptic transformations. We then express the corresponding quaternionic Möbius transformations in a "normal form" and show that, as

in $d = 1$, points converge to one of the fixed points for the hyperbolic class and oscillate for the elliptic class. This would lead to heating or non-heating phases in Floquet dynamics. We defer the discussion of behavior of physical quantities for large number of cycles to a future publication.

In the following, we calculate fidelities, unequal-time correlation functions and expectation values of the energy momentum tensor and primary operators in primary states at the end of a single drive cycle. Each cycle involves Hamiltonians of the form (1) with various $\Pi^\mu, \mu = 0, 1, \cdots 3$ for successive time intervals of equal value. The value of $\beta$ is either zero, or some fixed value, i.e. we do not consider different non-zero values of $\beta$ for different time intervals. These restrictions are for simplicity. All these diagonistics show that for $\beta < 1$ one has a non-heating phase with oscillatory behavior of these quantities as a function of time, while for $\beta > 1$ one has a heating phase characterized by exponential decays. $\beta = 1$ is a critical value, where the quantities have power law behavior. These three phases mirror the known dynamical phase structure in $1 + 1$ dimensions and reflect the three conjugacy classes of $SL(2, R)$. Since the driving Hamiltonians break rotation invariance on the $S^d$ the system develops inhomogenities. The nature of the inhomogeneties is richer in these higher dimensional cases, since we have the ability to break different sets of symmetries by choosing different sequences of $H_1 \cdots H_{d+1}$. Likewise, the structure of dynamical phases which can result from Floquet drives will be richer. It can be easily seen that if we use different nonzero values of $\beta$ the behavior is oscillatory when all the $\beta$ values are less than 1, power law when all the $\beta$ are equal to one. If any of the $\beta$ exceed 1, the behavior will be exponential.

The plan of the rest of this work is as follows. In Sec. 2 we study the properties of Hamiltonians of the form (1). This is followed by Sec. 3 which deals with our strategy for calculating the response to the dynamics for general $d$ and the formulation in terms of quaternions for $d = 3$. Next, in Sec. 4, we provide results for three quantities under the dynamics. The first corresponds to the fidelity of an evolving primary state at the end of a drive cycle; this is discussed in subsection (4.1). The second constitutes behavior of unequal-time correlation functions of the CFT under such evolution; this is discussed subsection(4.2). The evolution of the stress tensor, starting from a primary CFT state, is discussed in subsection (4.3). In section (4.4) we find the fixed points and fixed surfaces for the special class of quaternionic Möbius transformations mentioned above and determine the trajectories of points under successive cycles of a periodic drive. Finally, we discuss our main results and conclude in Sec. 5. Some details of the calculation are provided in the appendix.

## 2   The deformed CFTs

In this section we will discuss some properties of deformed CFT's with a Hamiltonian of the form (1).

### 2.1   $\beta < 1$ and Möbius quantization

When $\beta < 1$, for the choice $\Pi^\mu = (1, 0, 0, 0)$, the Hamiltonian $H$ (defined in (1)) can be thought of a different quantization of the theory with the undeformed hamitonian ($\beta = 0$) upto a scaling of the new time. For $d = 1$ this has been called "Möbius quantization".

Consider the Hamiltonian

$$H' = \frac{1}{\sqrt{1-\beta^2}} H = \cosh\theta\, 2iD + i\sinh\theta(K_0 + P_0) \qquad \beta = \tanh\theta \ . \tag{7}$$

This Hamiltonian is related to the dilatation operator by a similarity transformation

$$U^{-1}H'U \;=\; (2iD), \quad U = \exp\left[-\frac{i}{2}\theta(K_0 - P_0)\right] \ . \tag{8}$$

This can be verified by explicit calculation. However since the left hand side of (8) depends only on the commutators of the generators of the $SU(1,1)$ group, the equation must be independent of the specific representation.

To verify (8) it is convenient to use a representation of the $SU(1,1)$ using Pauli matrices,

$$D \;=\; i\sigma_z/2, \quad K_0 = \sigma_-, \quad P_0 = \sigma_+ \quad \sigma_\pm = \frac{1}{2}(\sigma_x \pm i\sigma_y). \tag{9}$$

The transformed special conformal and translation generators can be now deduced using the identities

$$\begin{aligned}
U^{-1}\sigma_z U &\equiv \tau_z = \sigma_z\cosh\theta + i\sigma_x\sinh\theta \\
U^{-1}\sigma_x U &\equiv \tau_x = \sigma_x\cosh\theta - i\sigma_z\sinh\theta, \quad \tau_y = \sigma_y \ .
\end{aligned} \tag{10}$$

Using the representation (9) this leads to

$$\begin{aligned}
D' &= D\cosh\theta + \frac{1}{2}(K_0 + P_0)\sinh\theta \\
K_0'[P_0'] &= \frac{1}{2}(1 + \cosh\theta)K_0[P_0] - \frac{1}{2}(1 - \cosh\theta)P_0[K_0] - D\sinh\theta \ .
\end{aligned} \tag{11}$$

To calculate the transformation of the other generators of the conformal algebra, it is useful to consider the combinations

$$A_\mu \equiv \frac{1}{2}(K_\mu - P_\mu) \qquad B_\mu \equiv \frac{1}{2}(K_\mu + P_\mu) \ . \tag{12}$$

The conformal algebra then leads to a closed $SL(2,R)$ subalgebra for the generators $(A_0, A_j, L_{0j})$ for each value of $j = 1, 2, 3$

$$[A_0, L_{0j}] = iA_j \qquad [A_0, A_j] = iL_{0j} \qquad [L_{0j}, A_j] = iA_0 \ . \tag{13}$$

Now consider the transformed $A_j$ or $L_{0j}$ (these angular momentum generators are defined in (5)),

$$A_j' = e^{i\theta A_0} A_j e^{-i\theta A_0} \qquad L_{0j}' = e^{i\theta A_0} L_{0j} e^{-i\theta A_0} \tag{14}$$

Once again, the result is determined entirely in terms of commutators. We can therefore use any representation of the algebra (13) to perform the calculation, e.g. the representation in terms of Pauli matrices

$$L_{0j} \to \frac{1}{2}\sigma_x \quad A_j \to \frac{1}{2}i\sigma_z \quad A_0 \to -\frac{1}{2}i\sigma_y \tag{15}$$

to obtain

$$\begin{aligned}
A_j' &= A_j\cosh\theta - L_{0j}\sinh\theta \\
L_{0j}' &= L_{0j}\cosh\theta - A_j\sinh\theta \ .
\end{aligned} \tag{16}$$

Furthermore the conformal algebra implies

$$[A_0, K_j + P_j] = [A_0, L_{ij}] = 0 \ . \tag{17}$$

This leads to final form of the deformed generators:

$$
\begin{aligned}
D' &= D \cosh\theta - \frac{1}{2}(K_0 + P_0)\sinh\theta \\
K_0' &= \frac{1}{2}(1 + \cosh\theta)K_0 - \frac{1}{2}(1 - \cosh\theta)P_0 - D\sinh\theta \\
P_0' &= \frac{1}{2}(1 + \cosh\theta)P_0 - \frac{1}{2}(1 - \cosh\theta)K_0 - D\sinh\theta \\
K_j' &= \frac{1}{2}(1 + \cosh\theta)K_j + \frac{1}{2}(1 - \cosh\theta)P_j - L_{0j}\sinh\theta \\
P_j' &= \frac{1}{2}(1 + \cosh\theta)P_j + \frac{1}{2}(1 - \cosh\theta)K_j + L_{0j}\sinh\theta \\
L_{0j}' &= \cosh\theta\, L_{0j} + \frac{1}{2}\sinh\theta(P_j - K_j), \quad L_{ij}' = L_{ij} \ .
\end{aligned}
\tag{18}
$$

These relations can be alternatively derived by first looking at the coordinate transformations resulting from the action of $U$ with infinitesimal $\theta$, and exponentiating them and requiring that the correct commutation relations are satisfied by the deformed generators. This is detailed in the appendix.

The results of this subsection imply that for $\beta < 1$ the deformed Hamiltonian is proportional to a standard CFT Hamiltonian (dilatation operator) which is quantized with a different notion of time. Consequently aspects of the physics of the deformed Hamiltonian are expected to be qualitatively similar to the undeformed Hamiltonian.

## 2.2 $\beta > 1$ and Lüscher-Mack Hamiltonians

The unitary transformation which relates the deformed theory to the undeformed theory for $\beta < 1$ does not work when $\beta > 1$. We will now demonstrate that for $\beta > 1$ the Hamiltonian can be instead deformed to the generator $K_0 + P_0$. Consider the Hamiltonian

$$H'' = \frac{1}{\sqrt{\beta^2 - 1}} H = 2iD\sinh\phi + i(K_0 + P_0)\cosh\phi \quad \beta = \coth\phi \ . \tag{19}$$

Using manipulations entirely similar to the previous subsection it is easy to show that

$$U^{-1} H'' U = i(K_0 + P_0), \quad U = \exp\left[-\frac{i}{2}\phi(K_0 - P_0)\right] \ . \tag{20}$$

The transformations of the other generators can be worked out following a procedure entirely analogous to the previous subsection. This shows that the physics of the deformed theory for $\beta > 1$ is similar to a theory with a Lüscher-Mack Hamiltonian ($K_0 + P_0$) rather than the dilatation operator.

As we will see, this difference manifests in dynamical processes like quantum quench by the appearance of a heating phase for $\beta > 1$ and an oscillating phase for $\beta < 1$.

# 3 Dynamics as a Conformal Transformation

As mentioned above, it is convenient to think of the time evolution by Hamiltonians like (1) by first performing a Weyl transformation to $R^{d+1}$ using Eqs. 3 and 4. Euclidean time evolution with the Hamiltonian $H(\beta, \Pi)$ is equivalent to a conformal transformation. This transformation can be obtained by using the Baker-Campbell-Hausdorff formula [25]

$$
\begin{aligned}
U_\mu \Pi^\mu \equiv U(\Pi) &= e^{-(2iwD + iw\beta(P_\mu + K_\mu)\Pi^\mu)} = e^{\Lambda_+ K_\mu \Pi^\mu} e^{\ln \Lambda_0 iD/2} e^{\Lambda_- P_\mu \Pi^\mu} \\
\Lambda_0 &= \left(\cosh w\nu_0 - (w\nu_0)^{-1}\sinh w\nu_0\right)^{-2}, \quad \nu_0 = \sqrt{1-\beta^2}, \\
\Lambda_+ &= -\Lambda_- = i\beta\nu_0^{-1}\sinh(w\nu_0)\Lambda_0^{1/2}
\end{aligned}
\tag{21}
$$

To write down the explicit transformations, we separate out the components. For example, using the above relations, one obtains:

$$
\begin{aligned}
e^{\alpha_1 iD} x_\nu e^{-\alpha_1 iD} &= e^{\alpha_1} x_\nu, \quad e^{i\alpha_2 P_\mu} x_\nu e^{-i\alpha_2 P_\mu} = x_\nu, \\
e^{i\alpha_3 K_\mu} x_\nu e^{-i\alpha_3 K_\mu} &= \frac{x_\nu}{1 - 2\alpha_3 x_\mu + r^2 \alpha_3^2},
\end{aligned}
\tag{22}
$$

where $K_\mu$ and $x_\mu$ denote the $\mu$-th component of the corresponding vectors.[2] Identifying $\alpha_1 = (1/2)\ln\Lambda_0$, $\alpha_2 = -i\Lambda_-$, and $\alpha_3 = -i\Lambda_+$, we find, after a few lines of algebra and for $\nu \neq \mu$

$$
x'_\nu = \frac{x_\nu}{\mathcal{D}_\mu},
\tag{23}
$$

where the denominator, $\mathcal{D}_\mu = \left[ \left(\cosh w\nu_0 - (w\nu_0)^{-1}\sinh w\nu_0\right)^2 + 2x_\mu \beta\nu_0^{-1}\sinh w\nu_0 \right.$

$$
\times \left. \left(\cosh w\nu_0 - (w\nu_0)^{-1}\sinh w\nu_0\right) + \beta^2 r^2 (w\nu_0)^{-2}\sinh^2 \tau\nu_0 \right].
$$

A similar, but more complicated expression can be obtained for $\nu = \mu$ in a similar manner. One obtains

$$
\begin{aligned}
x'_\mu &= \left[ x_\mu(2(\cosh^2 w\nu_0 - (w\nu_0)^{-2}\sinh^2 w\nu_0) - 1) - \beta\nu_0^{-1}\sinh w\nu_0(\cosh w\nu_0(1-r^2) \right. \\
&\quad \left. -(w\nu_0)^{-1}\sinh w\nu_0(1+r^2)) \right]/\mathcal{D}_\mu .
\end{aligned}
\tag{24}
$$

These finite conformal transformations can be then used to express time evolved quantities in terms of the quantities at initial time for any $d$.

## 3.1   $d = 3$ and quaternions

In $d \leq 3$ dimensions in general an efficient way to compute the transformed coordinates under the $SU(1,1)$ subgroup of conformal transformations is to use the quaternion formulation. In what follows, we provide details this formulation for $d = 3 + 1$. In Euclidean signature the coordinates are denoted by $X^\mu$ with $x_0 = \tau$, $x_1 = x$ and so on. We shall also define the $2 \times 2$ matrices $\tau_0 = I$ and $\tau_j = -i\sigma_j$ where $\sigma_j$ are the standard Pauli matrices and $I$ denotes the identity matrix. The first step of using the quaternion formulation is

---

[2]To present these formulae in a completely covariant form, one needs to implement an appropriate projection operator, which we do not use here. Our notation therefore breaks covariance. Nonetheless, we hope it is clear from the context whether we are referring to a vector or a particular component of it.

then to write the coordinates $x_\mu$ using a $2 \times 2$ matrix by associating each component of the coordinate to one of the $\tau_\mu$, where $\vec{\tau} = (I, -i\sigma_j)$, for $j = 1, 2, 3$:

$$Q_\nu = I x_\nu - i \sum_{j=1}^{3} \sigma_j y_j . \tag{25}$$

The $y_j$ which appears here are the components of $x^\mu$ with $\mu \neq \nu$. The choice of $\nu$ and $\mu$ is arbitrary at this stage. As an example, we may choose $\nu = 1$. The three other coordinates $x^j$ which appear are $y_1 = x_0 = \tau, y_2 = x_2 = y, y_3 = x_3 = z$, leading to

$$Q_1 = \begin{pmatrix} x - iz & -i(\tau - iy) \\ -i(\tau + iy) & x + iz \end{pmatrix} \tag{26}$$

Clearly $Q_\nu$ is not unique even after we fix $\nu$ and associate $x_\nu$ with the identity matrix since there is freedom of associating the rest of the coordinates with other Pauli matrices in different ways. As we shall see, all such choices lead to identical results for final coordinates under the class of conformal transformations that we discuss.

The action of the SU(1,1) transformations is then obtained by representing the generators by Pauli matrices. The operator $U(\Pi^\mu)$ in (21) is then represented by the $2 \times 2$ matrix,

$$U(\Pi^\mu) = e^{-w(-\sigma_z + i\beta\sigma_x)} = \begin{pmatrix} a_1 & a_2 \\ a_3 & a_4 \end{pmatrix}$$

$$a_1 = \left(\cosh w\nu_0 + \nu_0^{-1} \sinh w\nu_0\right), \quad a_4 = \left(\cosh w\nu_0 - \nu_0^{-1} \sinh w\nu_0\right)$$

$$a_2 = a_3 = -i\beta\nu_0^{-1} \sinh w\nu_0 \tag{27}$$

with $a_1 a_4 - a_2 a_3 = 1$. To find the transformed coordinates, we note that the quaternion matrix $Q_\mu$ transforms, upon action of $U(\Pi^\mu)$ to $Q'_\mu$ given by [26–29]

$$Q'_\mu = I x'_\mu - i \sum_{j \neq \mu} \sigma_j x'_j = (a_1 Q_\mu - i a_2 I).(i a_3 Q_\mu + a_4 I)^{-1}$$

$$= \begin{pmatrix} \frac{x_\mu(2a_1 a_4 - 1) - i(a_2 a_4 - a_1 a_3 r^2) - i x_{\nu_1}}{a_4^2 + 2i a_3 a_4 x_\mu - a_3^2 r^2} & \frac{-i(x_{\nu_2} - i x_{\nu_3})}{a_4^2 + 2i a_3 a_4 x_\mu - a_3^2 r^2} \\ \frac{-i(x_{\nu_2} + i x_{\nu_3})}{a_4^2 + 2i a_3 a_4 x_\mu - a_3^2 r^2} & \frac{x_\mu(2a_1 a_4 - 1) - i(a_2 a_4 - a_1 a_3 r^2) + i x_{\nu_1}}{a_4^2 + 2i a_3 a_4 x_\mu - a_3^2 r^2} \end{pmatrix} \tag{28}$$

where $\mu \neq \nu_1$, $\nu_2$, and $\nu_3$. The transformed coordinates are then obtained from the relation $x'_\alpha = \text{Tr}[\tau_\alpha^{-1} Q'_\mu]/2$ and yields

$$x'_\mu = \frac{x_\mu(2a_1 a_4 - 1) - i(a_2 a_4 - a_1 a_3 r^2)}{a_4^2 + 2i a_3 a_4 x_\mu - a_3^2 r^2}, \quad x'_\nu = \frac{x_\nu}{a_4^2 + 2i a_3 a_4 x_\mu - a_3^2 r^2} \tag{29}$$

Substituting Eq. 27 in Eq. 29, one recovers Eqs. 23 and 24. This shows that the quaternionic approach provides a vastly simpler way to obtain the transformed coordinates.

In the rest of the paper we will deal with $d = 3$. The transformation from $R \times S^3$ to $R^4$ is given by

$$\tau = e^w \cos\theta \ , x = e^w \sin\theta \cos\phi \ , y = e^w \sin\theta \sin\phi \cos\psi \ , z = e^w \sin\theta \sin\phi \sin\psi \ , \tag{30}$$

where $w_\mu = (w, \theta, \phi, \psi)$ are the coordinates on the cylinder $R \times S^3$, and $w$ denotes time on the cylinder.

While we have illustrated this formalism for $d = 3$ the same formalism can be used for $d = 1, 2$ by setting some of the coordinates in Eq.(26) to zero.

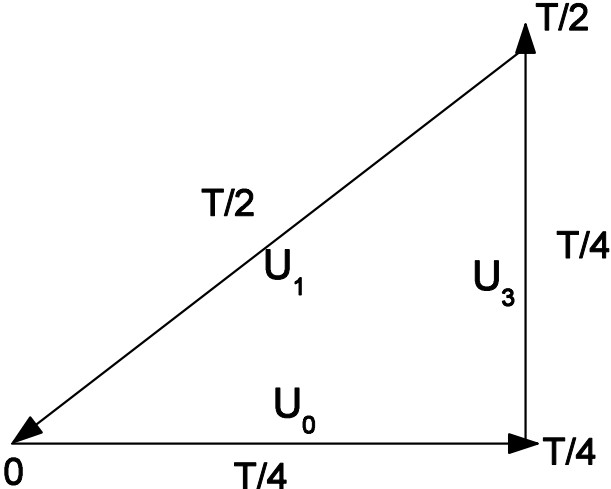

Figure 1: Schematic representation of the protocol for computing fidelity $F(t)$. See text for details.

## 4 Floquet Dynamics

In this section, we shall study Floquet dynamics of $3+1$ dimensional CFTs, using a square pulse protocol, described below. We will compute various physical properties at the end of each cycle. The strategy can be generalized to arbitrary number of cycles. We defer the analysis for multiple cycles to future work. In what follows, we shall choose the amplitude of the pulses within a drive period (denoted by $\beta$ below) to be same for simplicity.

The strategy will be to perform the computations on $R^4$, (coordinates $x^\mu$) Weyl transforming to $R \times S^3$ (coordinates $w^\mu$) and finally analytically continuing to real time $t = iw$. Under the Weyl transformation, a primary operator with conformal dimension $\Delta$ transforms as

$$O_\Delta(w_\mu) = e^{w\Delta}O_\Delta(x_\mu). \tag{31}$$

We shall use these relations to transform between the physical coordinates $w_\mu$ on the cylinder and those on the plane $(x_\mu)$.

As discussed above, unitary time evolution governed by a Hamiltonian of the form (1) is equivalent to a conformal transformation $x_\mu \to x'_\mu$ given by (31). The transformation of the primary operators of the CFT having conformal dimension $\Delta$ due to such dynamics is given by [30] :

$$O(x_\mu) \rightarrow U^\dagger O(x_\mu)U = O(x'_\mu)J_2^{\Delta/4}, \quad J_2 = \left|\frac{\partial x'_\mu}{\partial x_\nu}\right| = |\text{Det}[ia_3Q + a_4I]|^{-4} \tag{32}$$

where the last expression holds if the coordinates $\mathbf{x}'$ and $\mathbf{x}$ are related by the transformation given by Eqs. 27 and 28.

## 4.1  Fidelity

In this section, we shall compute the fidelity $F(T)$ of a primary state $|\Delta\rangle$ of the CFT at the end of a drive cycle. This is defined, in Euclidean time, as

$$F(T_0) \quad = \quad \frac{\langle\Delta|U(T_0,0)|\Delta\rangle}{\langle h|h\rangle} = \frac{\lim_{x_{2\mu}\to\infty, x_{1\mu}\to 0}\langle 0|\phi(\mathbf{x}_2)U(T_0,0)\phi(\mathbf{x}_1)|0\rangle}{\lim_{x_{2\mu}\to\infty, x_{1\mu}\to 0}\langle 0|\phi(\mathbf{x}_2)\phi(\mathbf{x}_1)|0\rangle} \tag{33}$$

where $\phi(\mathbf{x}) \equiv \phi(\tau, x, y, z)$ denotes a primary CFT field of dimension $h$, $T = -iT_0$ is the drive period in real time and $|0\rangle$ denotes the CFT vacuum. Throughout this section, we shall work in Euclidean time and analytically continue to real time whenever necessary.

The protocol we choose for computing $F(T)$ is schematically shown in Fig. 1. The evolution operator $U(\Pi^\mu)$ in each cycle is chosen to be

$$U(\Pi^\mu; \tau, 0) \quad = \quad e^{-H(\Pi^\mu)\tau}, \quad H(\Pi^\mu) = i(2D + \beta(K_\mu + P_\mu)\Pi^\mu \ . \tag{34}$$

The total evolution operator at the end of one cycle is given by

$$U = U^\dagger(\Pi^1; 0, T_0/2)U(\Pi^3; T_0/2, T_0/4)U(\Pi^0; T_0/4, 0) \ , \tag{35}$$
$$\Pi^1 = (0, 1, 0, 0) \ , \quad \Pi^3 = (0, 0, 0, 1) \ , \quad \Pi^0 = (1, 0, 0, 0) \ . \tag{36}$$

Note that we use the different deformed CFT Hamiltonians with same deformation parameter $\beta$ to generate the evolution operator $U$. The fidelity is computed at the end of the cycle.

To obtain $F(T)$, we first note that the two point correlation function of a primary operator with dimension $\Delta$ can be written in the quaternion formalism as [28]

$$C_0 \quad = \quad \langle 0|\phi(\mathbf{x}_2)\phi(\mathbf{x}_1)|0\rangle = \frac{1}{(\text{Det}[Q(\mathbf{x}_2) - Q(\mathbf{x}_1)])^\Delta} \tag{37}$$

In the above equation we have not specified a subscript for the quaternions since this particular result is independent of which $Q_{(\mu)}$ we use. For the unequal-time correlation function under a transformation by $U_\mu(T, 0)$, this leads to

$$\begin{aligned} C_1 \quad &= \quad \langle 0|U_\mu^\dagger(T, 0)\Pi^\mu\phi(\mathbf{x}_2)U_\mu(T, 0)\Pi^\mu\phi(\mathbf{x}_1)|0\rangle \\ &= \quad \frac{1}{\text{Det}[Q'_\mu(\mathbf{x}'_2) - Q_\mu(\mathbf{x}_1)]^\Delta}\text{Det}[(ia_3 Q_\mu(\mathbf{x}_2) + a_4 I)^{-1}]^\Delta \\ &= \quad \frac{1}{\text{Det}[(a_1 Q_\mu(\mathbf{x}_2) - ia_2 I) - Q_\mu(\mathbf{x}_1)(ia_3 Q_\mu(\mathbf{x}_2) + a_4 I)]^\Delta}. \end{aligned} \tag{38}$$

Note that the second term in the denominator vanishes when $x_{1\mu} \to 0$. Further when $x_{2\mu} \to \infty$, we find $C_1 \to 1/\text{Det}[a_1 Q_{\mu_1}(\mathbf{x}_2)]^\Delta$. Furthermore similar analysis shows for multiple subsequent transformations given by

$$U_{\mu_i}\Pi^\mu \quad = \quad \begin{pmatrix} a_i & b_i \\ c_i & d_i \end{pmatrix}, \quad a_i d_i - b_i c_i = 1 \tag{39}$$

$C_1 \to 1/\text{Det}[(\prod_i a_i)Q_{\mu_i}(\mathbf{x}_2)]^\Delta$. This allows us to write the final expression for $F(T_0)$

$$F(T_0) \quad = \quad \frac{1}{(a_0(T_0)a_3(T_0)a_1(T_0))^\Delta} \tag{40}$$

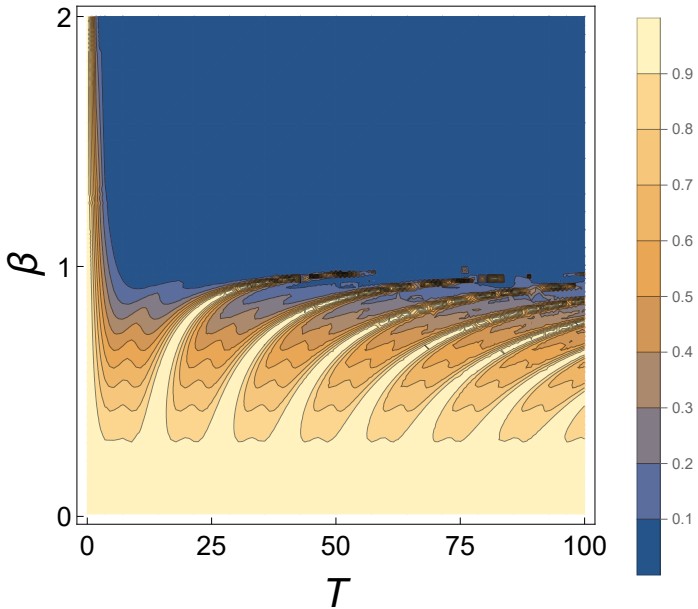

Figure 2: Plot of $|F(T)|$ as a function of $\beta$ and $T$ with $\Delta = 1$. For $\beta < 1$, $|F(T)|$ oscillates with $T$ characterizing the non-heating phase and reaches $\simeq 1$ for $\nu = 4n\pi$ where $n$ is an integer. For $\beta > 1$, $|F(T)|$ decays exponentially with $T$ which is a signature of the heating phase.

To compute the $a_0$, $a_1$ and $a_3$, we use the $2 \times 2$ matrix representation of the operators $D$, $K_\mu$ and $P_\mu$ given in Eq. 9. Using this one can write

$$U_\mu(\tau)\Pi^\mu = e^{-\tau\sqrt{1-\beta^2}(-n_z\sigma_z + n_x\sigma_x)}, \quad n_z = \frac{1}{\sqrt{1-\beta^2}}, \quad n_x = \frac{i\beta}{\sqrt{1-\beta^2}} \tag{41}$$

Defining $\nu = \sqrt{1-\beta^2}\,T/4$, we find, after analytically continuing to real time $T = -iT_0$,

$$a_0(T) = a_3(T) = (\cos\nu + iT(4\nu)^{-1}\sin\nu), \quad a_1(T) = (\cos 2\nu - iT(2\nu)^{-1}\sin 2\nu)$$

$$F(T) = \frac{1}{[(\cos\nu + iT(4\nu)^{-1}\sin\nu)^2(\cos 2\nu - iT(2\nu)^{-1}\sin 2\nu)]^\Delta} \quad \text{for } \beta \neq 1$$

$$= \left(\frac{1}{(1 + iT/4)^2(1 - iT/2)}\right)^\Delta \quad \text{for } \beta = 1 \tag{42}$$

The behavior of each of the $a_i$ and hence $F(T)$ depends crucially on the value of the parameter $\beta$. For $\beta < 1$ these are oscillatory functions of the time $T$, while for $\beta > 1$ they decay exponentially for large $T$. These two behaviors are higher dimensional versions of the non-heating and heating phases. In between these phases there is a critical point $\beta = 1$ where we have a power law decay in time, $|F(T)| \sim 1/[(1 + T^2/16)^2(1 + T^2/4)]^{\Delta/2}$. This behavior is charted out in Fig 2 which shows the behavior $|F(T)|$ for $\Delta = 1$. In between, at $\beta = 1$, we find a line which represents a critical line separating the two phases.

Our analysis of $F(T)$ indicates special frequencies at which $|F(T)| = 1$ indicating perfect overlap of the driven state with the initial states. These frequencies, which exist only in the non-heating phase can be read off from Eq. 42 and are given by $T = T_n^*(\beta)$ where

$$T_n^*(\beta) = 4n\pi/\sqrt{1-\beta^2}, \tag{43}$$

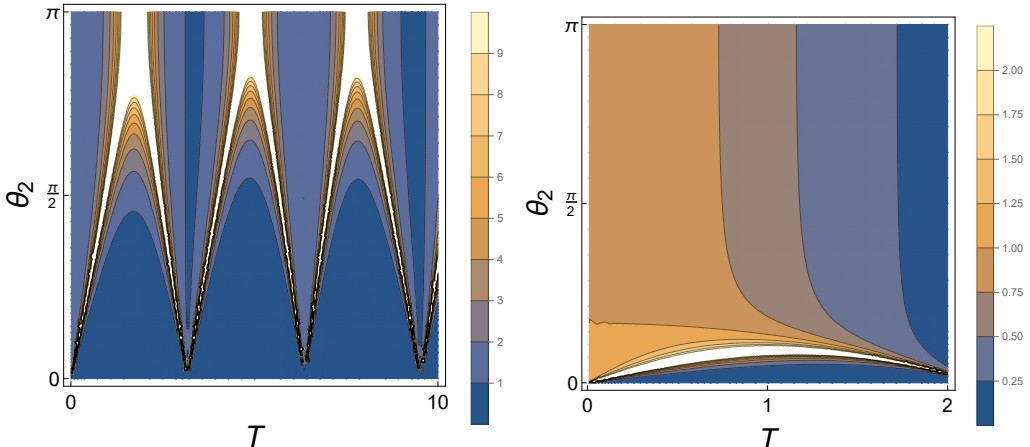

Figure 3: Plot of $|C_1(T)/C_1(0)|$ as a function of $\theta_2$ and $T$ for $\beta = 0.2$ (left panel) and $\beta_1 = 1.2$ (right panel) corresponding to a square pulse protocol. See text for details.

and $n$ is an integer. The existence of these frequencies can also be seen in Fig. 2.

Thus we find that under a drive schematically represented by Fig. 1, $F$ exhibits perfect revival at special frequencies in the non-heating phase and an exponential (power-law) decay with $T$ in the heating phase (on the critical line).

The different behaviors for different values of $\beta$ reflect the properties of the Hamiltonians used for time evolutions which have been discussed in the previous section. Each of these Hamiltonians represent a *different* $SU(1,1)$ subgroup of the conformal group. We chose the same $\beta$ for each of the three Hamiltonians, so the conjugacy classes of each of these $SU(1,1)$ are the same. The results can be trivially extended to the cases where the three $\beta$'s are different.

## 4.2    Unequal-time Correlators

In this section, we compute the unequal-time two-point correlation function of the primary fields with conformal dimension $\Delta$ in the presence of a drive. We consider the fields on a cylinder and map them onto the plane using Eq. 31. The initial coordinates on the plane corresponds to

$$
\begin{aligned}
\mathbf{x_2} &= (\tau_2, x_2, y_2, z_2) = (\cos\theta_2, \sin\theta_2\cos\phi_2, \sin\theta_2\sin\phi_2\cos\psi_2, \sin\theta_2\sin\phi_2\sin\psi_2), \\
\mathbf{x_1} &= (\tau_1, x_1, y_1, z_1) = (1, 0, 0, 0)
\end{aligned}
\tag{44}
$$

where we have taken initial time on the cylinder $w = 0$ without loss of generality. We have also chosen the coordinates $\theta_1 = 0$ for simplicity. In what follows we shall consider the correlation function

$$
C_1(T) = \langle 0|U^\dagger(T,0)O(\mathbf{x_2})U(T,0)O(\mathbf{x_1})|0\rangle = \frac{J_2^{\Delta/4}}{(\mathrm{Det}[Q(\mathbf{x_2'}) - Q(\mathbf{x_1})])^\Delta}
\tag{45}
$$

where $\mathbf{x_2'}$ represents the transformed coordinate and $J_2$ denotes the Jacobian of the coordinate transformation.

To study the dynamics, we first consider a square pulse protocol given by

$$
\begin{aligned}
H &= H_{(-)} = 2iD \qquad \text{for } t \leq T/2 \\
H &= H_{(+)} = 2iD + i\beta(K_0 + P_0) \qquad \text{for } t > T/2.
\end{aligned} \tag{46}
$$

We will calculate the correlator at time $T$.

The evolution operator for this is

$$
U(T,0) = U_0(T,0) = e^{-T_0(2iD + i\beta(K_0+P_0))/2} e^{-iDT_0} = \begin{pmatrix} a & b \\ c & d \end{pmatrix}
$$

$$
a = d^* = \left( \cosh \eta T/2 + \frac{i}{\eta} \sinh \eta T/2 \right) e^{iT/2}, \quad b = c^* = e^{-iT/2} \frac{\beta}{\eta} \sinh \eta T/2, \tag{47}
$$

$$
Q_0(\mathbf{x}_2') = (aQ_0(\mathbf{x_2}) - ibI).(icQ_0(\mathbf{x_2}) + dI)^{-1}, \quad J_2 = |(\text{Det}[icQ_0(\mathbf{x_2}) + dI]^\Delta|^{-4} \tag{48}
$$

where $\eta = \sqrt{\beta^2 - 1}$ and $I$ is the $2 \times 2$ identity matrix. Here we have used Eqs. 27 and 29, and performed a Wick rotation $T = -iT_0$. Substituting Eqs. 48 in Eq. 45, we find after transforming back to cylinder coordinates [3]

$$
\begin{aligned}
C_1(T) &= C_1(0) \left[ \frac{2(1 - \cos\theta_2)}{(a - ic)^2 - (b - id)^2 - 2i(a - ic)(b - id)\cos\theta_2} \right]^\Delta, \\
C_1(0) &= \frac{1}{2(1 - \cos\theta_2)^\Delta}
\end{aligned} \tag{49}
$$

The dependence of $C_1(T)$ on both $T$ and $\beta$ can be obtained by substituting the expressions of $a$, $b$, $c$ and $d$ from Eq. 48. We note that for $\beta > 1$, the system is in the heating phase and the correlation decays exponentially; in contrast, for $\beta < 1$, the dynamics is oscillatory. Also, we find that $C(T)$ depends only on $\theta_2$ and is independent of other coordinates. This is a consequence of the choice of the drive protocol which involves only $K_0$ and $P_0$. These features of the correlation function are shown in Fig. 3 where $|C_1(T)/C_1(0)|$ is plotted for $\Delta = 1$ as a function of $\theta_2$ and $T$ for $\beta = 0.2$ (left panel) and $\beta = 1.2$ (right panel).

We next consider a drive protocol involving two different sets of generators given by

$$
H_{(-)} = 2iD + i\beta(K_{(0)} + P_{(0)}), \quad H_{(+)} = 2iD + i\beta(K_{(3)} + P_{(3)})
$$

$$
U(T_0,0) = U_{(+)}(T_0, T_0/2) U_{(-)}(T_0/2, 0), \quad U_{(\pm)}(\tau, 0) = e^{-\tau H_{(\pm)}/\hbar} = \begin{pmatrix} a_\pm & b_\pm \\ c_\pm & d_\pm \end{pmatrix} \tag{50}
$$

To obtain the correlator corresponding to this protocol, we consider the action of $U_-$ on $Q(\mathbf{x}_2) \equiv Q_2$ which is given by

$$
\begin{aligned}
Q_2^{(1)} &= (a_- Q_2 - ib_- I).(ic_- Q_2 + d_- I)^{-1} = \begin{pmatrix} \tau' - iz' & -i(x' - iy') \\ -i(x' + iy') & \tau' + iz' \end{pmatrix} \\
\tau' &= \frac{\tau(2a_- d_- - 1) - i(a_- c_- - b_- d_- r^2)}{d_-^2 + 2ic_- d_- \tau - c_-^2 r^2}, \quad x_j' = \frac{x_j}{d_-^2 + 2ic_- d_- \tau - c_-^2 r^2}
\end{aligned} \tag{51}
$$

where $x_j = (x, y, z)$ for $j = (1, 2, 3)$. Next, we rewrite $Q_2^{(1)}$ as $Q_2'^{(1)} = z'I - -i\sigma_z \tau' - i\sigma_x x' - i\sigma_y y'$ and perform the second transformation

$$
\begin{aligned}
Q_2^{(2)} &= (a_+ Q_2'^{(1)} - ib_+ I).(ic_+ Q_2'^{(1)} + d_+ I)^{-1} = \begin{pmatrix} z'' - i\tau'' & -i(x'' - iy'') \\ -i(x'' + iy'') & z'' + i\tau'' \end{pmatrix} \\
z'' &= \frac{z'(2a_+ d_+ - 1) - i(a_+ c_+ - b_+ d_+ r'^2)}{d_+^2 + 2ic_+ d_+ z' - c_+^2 r'^2}, \quad x_j'' = \frac{x_j'}{d_+^2 + 2ic_+ d_+ z' - c_+^2 r'^2}
\end{aligned} \tag{52}
$$

---

[3]The Weyl factors cancel in the following expressions

where $x_j'' = (\tau'', x'', y'')$ for $j = (1, 2, 3)$.

The rest of the calculation is cumbersome but straightforward. A somewhat lengthy algebra yields

$$C_2(T) = C_1(0) \left( \frac{2(1 - \cos\theta_2)}{\alpha + \beta_1 \cos\theta_2 + \beta_2 \sin\theta_2 + \beta_3 \cos 2\theta_2 + \gamma(\theta_2) \sin\theta_2 \sin\phi_2 \sin\psi_2} \right)^\Delta . \quad (53)$$

Here $\alpha$, $\beta_i$ $(i = 1, 2, 3)$, and $\gamma$ are functions of the drive parameters through functions $a$, $b$, $c$ and $d$ given by

$$
\begin{aligned}
a &= d^* = \left( \cosh\eta T/2 + \frac{i}{\eta}\sinh\eta T/2 \right), \quad b = c = \frac{\beta}{\eta}\sinh\eta T/2, & (54) \\
\gamma(\theta_2) &= -4ib(a - d)(b^2 - d^2 - 2ibd\cos\theta_2), \quad \beta_2 = b^2 - a^2 \\
\beta_3 &= \frac{1}{2}\Big[ 8a^3d(1 - ad) + a^2(-1 + 8(b + 2id)d^2) - 8iabd(d + b(2bd - i)) \\
&\quad + b^2(1 + 8d(ib - d(b^2 - d^2))) \Big] \\
\beta_1 &= 4i\Big[ 2a^4bd - c^3b(1 + 2d^2) + a(b^3 + ib^2d(1 + 4b^2) + 2b^3d^2 - id^3) - b^3d(1 + 4d^2) \\
&\quad + a^2d(b - 2b^3 - 4ib^2c4 + 2id^3) + b(2b^4d + ibd^2 + 2d^5 - ib^3(1 + 2d^2)) \Big]
\end{aligned}
$$

where we have analytically continued to real time $T = -iT_0$. The function $\alpha$ is cumbersome but can be expressed in terms of $a$, $b$, $c$ and $d$.

Eq. 53 indicates a clear rotational symmetry breaking in the correlation function. This is a consequence of application of two different sets of generators $(K_0, P_0)$ and $(K_3, P_3)$ for constructing $U(T, 0)$. This dependence is shown for a fixed $T = 1$, $\theta_2 = \pi/4$ and $\beta = 0.2$ in Fig. 4 where $|C_2(T)/C_2(0)|$ is plotted as a function of $\phi_2$ and $\psi_2$.

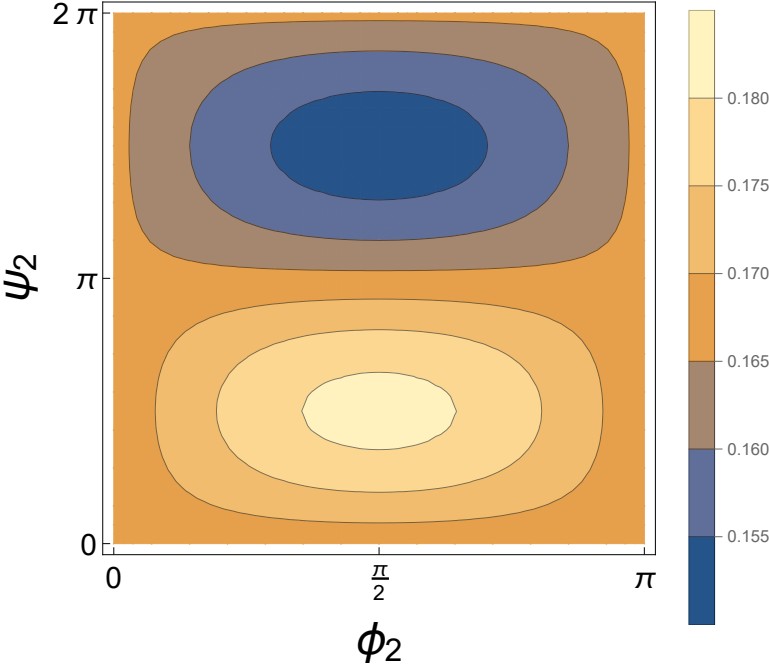

Figure 4: Plot of $|C_2(T)/C_2(0)|$ as a function of $\phi_2$ and $\psi_2$ for $\beta = 0.2$, $T = 5$ and $\theta = \pi/4$. See text for details.

### 4.3 Local probes of the driven state

In this section we will probe the time-evolving state with local operators, i.e., we are interested in computing :

$$\langle \Delta | U_\mu^\dagger(T,0)\Pi^\mu A(w^\alpha) U_\mu(T,0)\Pi^\mu|\Delta\rangle. \tag{55}$$

We will find here as in the $d=1$ case [31] that there is localization in the local observables.

**Energy density**

Consider first the energy density, i.e. when the operator $A(w^\alpha)$ is chosen to be $T_{ww}(w^\alpha)$. Therefore we will need the transformation of the stress tensor. For $d=3$, the conformal dimension of the energy-momentum tensor is 4, so that the Weyl factor is simply $e^{4w}$. Along with the factors coming from the transformation of a spin-2 tensor, the energy density translates to

$$T_{ww}(w^\mu) = e^{4w}\frac{\partial x^\rho}{\partial w}\frac{\partial x^\sigma}{\partial w}T_{\rho\sigma}(x) + \frac{3a}{8\pi^2}, \tag{56}$$

where the last term comes from the Weyl anomaly. Conjugation of the $R^{d+1}$ stress tensor $T_{\rho\sigma}(x)$ gets with the deformed evolution operators $U_\mu(T,0)\Pi^\mu$ leads to a conformal transformation. In $D$ flat spacetime for a spin-2 conformal primary like the stress-tensor the transformation rule is:

$$U^\dagger T_{\mu\nu}(x)U = T'_{\mu\nu}(x) = J^{\frac{D-2}{D}}\frac{\partial x'^\rho}{\partial x^\mu}\frac{\partial x'^\sigma}{\partial x^\nu}T_{\rho\sigma}(x'), \tag{57}$$

where $J$ denotes the Jacobian of the conformal transformation. Finally we shall be left to compute a plane three point function involving a stress-tensor and two primaries. This is given by [32]:

$$\langle O(x_1)O(x_2)T^{\mu\nu}(x_3)\rangle = -\frac{4\Delta}{6\pi^2}\frac{H^{\mu\nu}(x_1,x_2,x_3)}{|x_{12}|^{2\Delta-2}|x_{13}|^2|x_{23}|^2}, \tag{58}$$

$$\text{where, } H^{\mu\nu} = V^\mu V^\nu - \frac{1}{4}V^2\delta^{\mu\nu}, \text{ with, } V^\mu = \frac{x_{13}^\mu}{x_{13}^2} - \frac{x_{23}^\mu}{x_{23}^2}.$$

Putting everything together, the time-dependent piece (i.e. modulo the anomalous piece) in the energy density of the driven state is:

$$\langle \Delta|U^\dagger T_{ww}(w^\mu)U|\Delta\rangle_T = -2\frac{\Delta}{3\pi^2}\frac{e^{4w}J_2^{-1/2}}{|x'(w^\mu)|^2}\frac{\partial x^\rho}{\partial w}\frac{\partial x^\sigma}{\partial w}\frac{\partial x'^\alpha}{\partial x^\rho}\frac{\partial x'^\beta}{\partial x^\sigma}H_{\alpha\beta}(x'(w^\mu)), \tag{59}$$

$$\text{where, } H_{\rho\sigma}(x') = \frac{x'_\rho x'_\sigma - \frac{x'^2}{4}\delta_{\rho\sigma}}{(x')^4}.$$

We consider a square pulse drive given by:

$$
\begin{aligned}
H &= H_{(-)} = 2iD \quad\quad \text{for } t \le T/2\\
H &= H_{(+)} = 2iD + i\beta\left(K_{(1)} + P_{(1)}\right) \quad\quad \text{for } t > T/2.
\end{aligned} \tag{60}
$$

For this protocol the evolution operator is parametrized as : $\begin{pmatrix} a & b \\ c & d \end{pmatrix}$ with $a,b,c,d$ as in Eq. 47. Below in Fig. 5 and Fig. 6 we plot the absolute value of the normalized energy

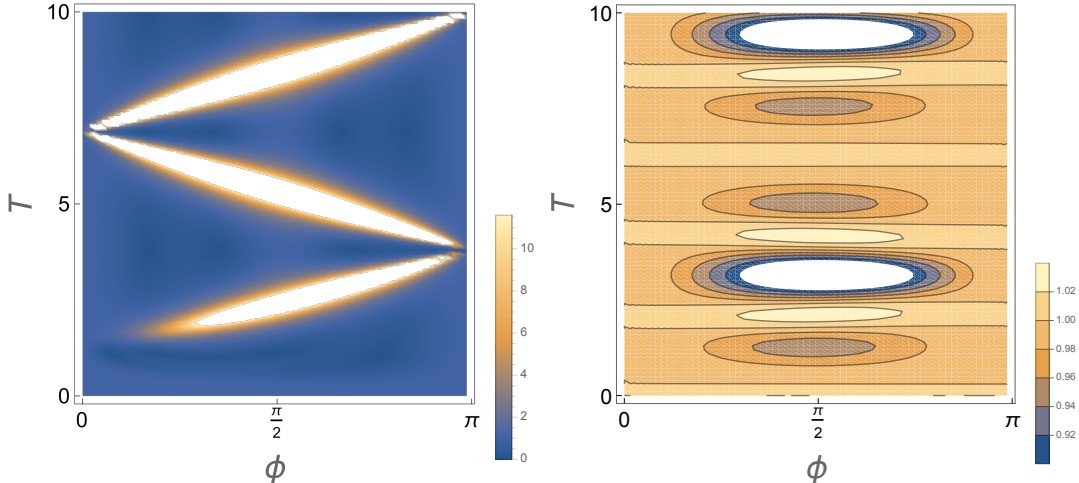

Figure 5: Density plots of normalized density $|E(\theta, \phi, T)|$ in the $\phi$, $T$ plane for $\theta = \pi/2$. Left Panel: In the heating phase with $\beta = 1.2$ we find progressive localization. Right panel: In the non-heating phase with $\beta = 0.09$ we find oscillations.

density

$$E(\theta, \phi, \psi, T) = \frac{\langle \Delta | U^\dagger T_{ww}(w^\mu) U | \Delta \rangle_T}{\langle \Delta | U^\dagger T_{ww}(w^\mu) U | \Delta \rangle_{T=0}} \tag{61}$$

in various regimes. Note that $\beta$ denotes the amplitude of the deformation and can be used to enter and leave the heating regime. When $\beta > 1$ and we are in the heating regime we find clear signatures of localization of the energy density in the angular directions, whereas the non-heating regime is characterized by oscillations. Furthermore the localization occurs in both the angular directions $\theta$ and $\phi$ as is clear from the density plot of Fig. 7, and is independent of the $\psi$ direction. Due to the latter feature we omit $\psi$ from $E(\theta, \phi, \psi, T)$.

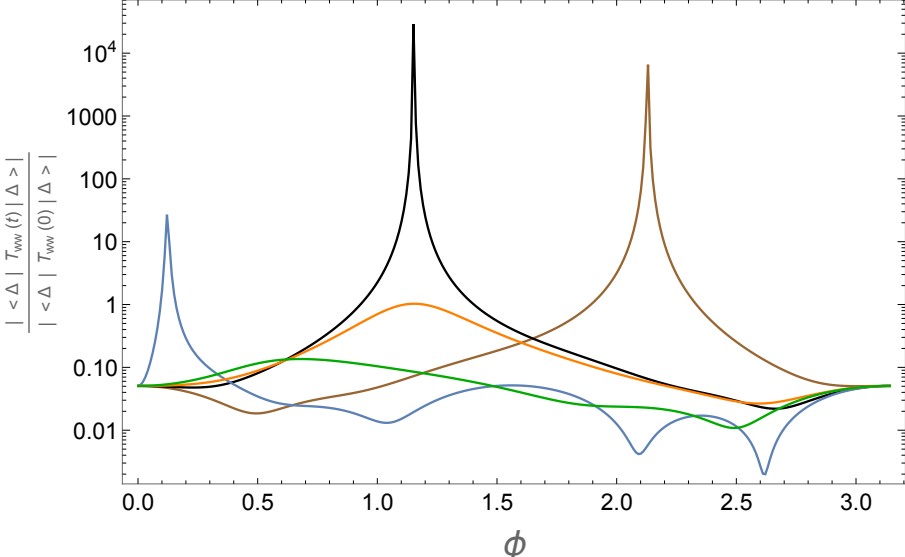

Figure 6: Plots of normalized density $|E(\theta, \phi, T)|$ as a function of $\phi$ for $\theta = \pi/2$ in the heating phase, $\beta = 1.2$, and for several representative values of $T$ : 1.5 (Green), 2 (Orange), 7 (Blue), 9 (Brown) and 12 (Black).

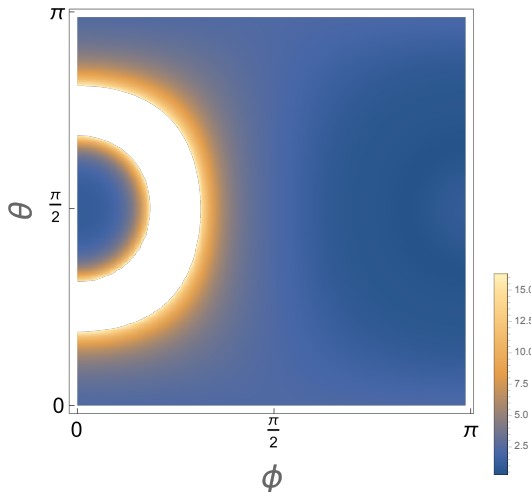

Figure 7: Density plots of normalized density $|E(\theta, \phi, T)|$ in the $\phi$, $\theta$ plane for $T = 6$, in the heating phase, $\beta = 1.1$.

The explicit formula for the energy density as obtained from the analytic expression Eq. 59 turns out to be too complicated. Thus we turn our attention to a simpler observable, namely, that of a local primary probe in the driven state, which allows for an explanation of the angular localization.

### Primaries

Here we choose the probe to be another primary operator in place of $A(w^\alpha)$ in Eq. 55. The transformation rule of a primary is already given as in Eq. 32, using which along with the Weyl factor, Eq. 31, and the universal formula for three point primary correlator, we obtain:

$$\langle \Delta | U_\mu^\dagger \Pi^\mu(T, 0) O_{\Delta_1}(w^\alpha) U_\mu(T, 0) \Pi^\mu | \Delta \rangle = e^{w\Delta_1} J_2^{\Delta_1/4} \frac{C_{\Delta\Delta\Delta_1}}{|x'|^{\Delta_1}}, \tag{62}$$

where $C_{\Delta\Delta\Delta_1}$ is the operator product expansion coefficient which is part of the CFT data. The above expression in the cylindrical coordinates, with initial time $w = 0$, for a generic $SU(1,1)$ drive in the $x$ direction, takes the form :

$$
\begin{aligned}
\langle \Delta | O_{\Delta_1}(T) | \Delta \rangle &= C_{\Delta\Delta\Delta_1} \frac{[((1 + a_2 a_3)^2 - a_2^2 a_4^2)^2 + 4a_2^2 a_4^2 (1 + a_2 a_3)^2 \cos^2 \phi \sin^2 \theta]^{-\Delta_1/4}}{a_4^{-4\Delta_1}[a_4^4(a_3^2 - a_4^2)^2 + 4a_3^2 a_4^6 \sin^2 \theta \cos^2 \phi]^{3\Delta_1/4}} \\
&= C_{\Delta\Delta\Delta_1} O(\theta, \phi, T)_\Delta.
\end{aligned}
\tag{63}
$$

Once again there is independence in $\psi$. Notice that when either $\theta = 0$ or $\phi = \pi/2$ the above becomes space independent. Once again, we consider the square-pulse protocol involving deformation in the $x$ direction, as in Eq. 60. The parameter $\beta$ denotes the amplitude of the drive and can be used to enter and leave the heating regime. In Fig.8 on the $\beta, T$ plane we obtain a density plot very similar to Fig.2. In particular we notice, that oscillations die into exponential fall-offs as the $\beta = 1$ line is crossed from below.

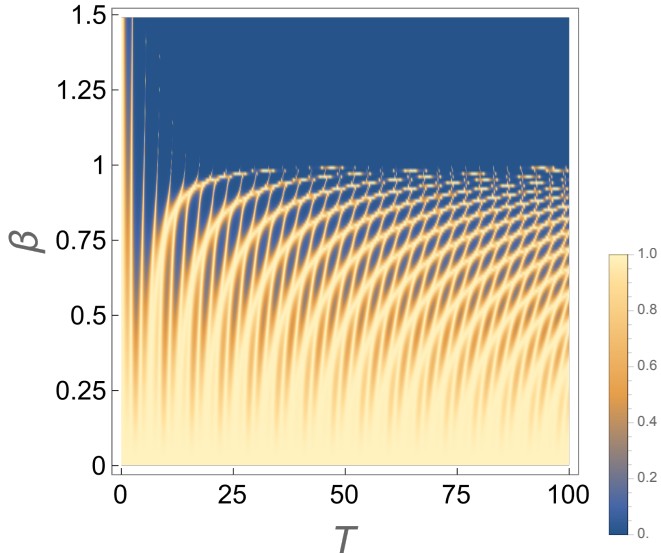

Figure 8: Plot of $|O(\theta = 0, \phi, T)|_\Delta$ as a function of $\beta$ and $T$ with $\Delta_1 = 1$. For $\beta < 1$, we find oscillations with $T$ characterizing the non-heating phase and for $\beta > 1$, the one-point function decays exponentially with $T$ which is a signature of the heating phase.

At $\beta = 1$ the functional dependence of the absolute value squared of the one point function in the driven state is :

$$|O(\theta, \phi, T)|_\Delta^2 = \Big( (8 + 4T^2 + T^4 + T^2(T^2 - 4)\cos 2T + 4T^3 \sin 2T)^2 - 4T^2 \cos^2 \phi (T^2(4 + T^2)^2$$

$$+ (8 + 4T^2 + T^4)((T^2 - 4)\cos 2T + 4T \sin 2T))\sin^2 \theta + 4T^4(4 + T^2)^2 \cos^4 \phi \sin^4 \theta \Big)^{-\Delta_1}.$$

$$(64)$$

At large $T$, dropping the rapidly oscillatory pieces, we find:

$$|O(\theta, \phi, T)|_\Delta^2 \sim \frac{1}{T^{8\Delta_1}(1 - 2\cos^2 \phi \sin^2 \theta)^{2\Delta_1}} + \mathcal{O}(T^{-10\Delta_1}). \tag{65}$$

It turns out that the higher order terms in the $1/T$ series (with rapidly oscillating terms dropped) contains higher order singularities of $1 - 2\cos^2 \phi \sin^2 \theta$ :

$$|O(\theta, \phi, T)|_\Delta^2 = \sum_{n=1}^{\infty} \frac{f_{n,\Delta_1}(\cos^2 \phi \sin^2 \theta)}{T^{\Delta_1(6+2n)}(1 - 2\cos^2 \phi \sin^2 \theta)^{\frac{1}{2}(2n+1-(-1)^n)\Delta_1}}. \tag{66}$$

These singularities give rise to localization in the angular directions, and as we numerically investigate next, also persists in the heating, $\beta > 1$ regime.

Below in Fig.9 we plot the normalized one point function amplitude in various regimes. We find clear signatures of localization of the amplitude in the angular directions, where as the non-heating regime is characterized by oscillations.

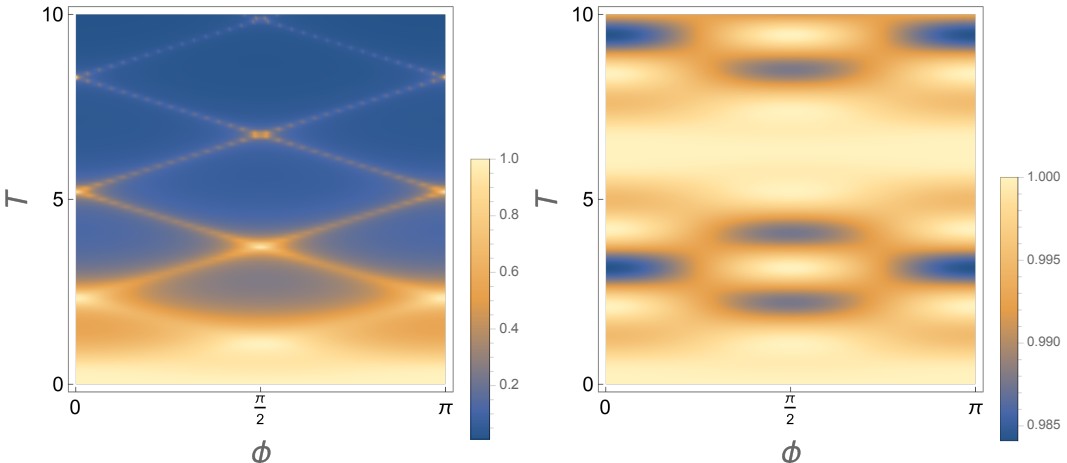

Figure 9: Density plots of normalized $|O(\theta, \phi, T)|_\Delta$ in the $\theta$, $T$ plane for $\phi = 0$. Left Panel: In the heating phase with $\beta = 1.1$ we find progressive localization. Right panel: In the non-heating phase with $\beta = 0.1$ we find oscillations. The conformal dimension is taken to be $\Delta_1 = 0.4$.

Localization occurs in both the $\phi$ as well as $\theta$ directions as is clear from the left panel of Fig.10. In the right panel of Fig.10 we have plotted the contours of the function $\cos^2 \phi \sin^2 \theta$ since at least in the $\beta = 1$ line we expect from Eq. (66) localization along the $\frac{1}{2}$ contour. The plots are indicative of the fact that similar singularities extend into the heating regime as well. It is very natural that a similar effect is responsible for the localization in the energy density when the primary gets replaced by the stress-tensor.

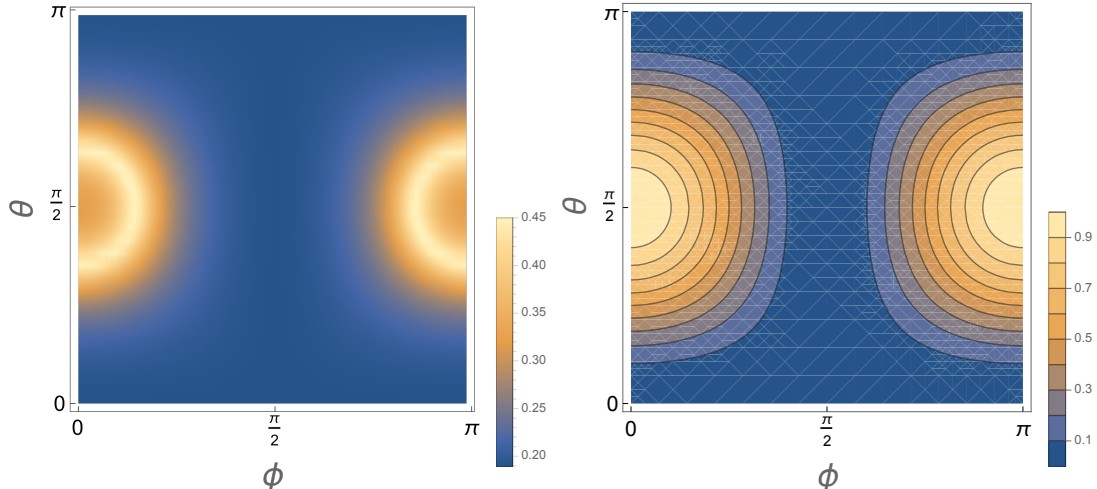

Figure 10: Left Panel : Localization from the density plots of normalized $|O(\theta, \phi, T)|_\Delta$ in the $\phi$, $\theta$ plane for $T = 3.3$, in the heating phase, $\beta = 1.1$. The conformal dimension is set to $\Delta_1 = 0.4$. Right Panel : Contour lines of $\cos^2 \phi \sin^2 \theta$. See text for details.

## 4.4 Multiple cycles and dynamical phases

In the sections above we have presented the results for a single cycle of a periodic protocol. However the formalism developed above can be recursively applied to multiple cycles, and Floquet dynamics as a function of the stroboscopic time can be studied. By varying the time intervals we expect to observe dynamical phases and transitions between them.

When a cycle contains Hamiltonians belonging to one $SL(2, R)$ algebra, we can judiciously choose quaternionic representation of the coordinates on $R^4$, such that the parameters of the quaternionic Möbius transformation $SL(2, H)$ are real, and the discussion of multiple cycles becomes simple.

Consider a general conformal transformation (we omit the index $\mu$ for the quaternion $Q_\mu$ below)

$$Q' = (aQ + b) \cdot (cQ + d)^{-1} \tag{67}$$

where $a, b, c, d$ are themselves quaternions, satisfying

$$\text{Det}|ac^{-1}dc - bc|^2 = 1 \tag{68}$$

For transformations which involve the subalgebra formed by $D, K_0, P_0$, one can choose

$$Q = x_0 I - i \sum_{i=1}^{3} \sigma_i x_i \tag{69}$$

Then for such transformations $a, b, c, d$ are real numbers satisfying

$$ad - bc = 1 \tag{70}$$

A fixed point $\bar{Q}$ satisfies

$$\bar{Q}(c\bar{Q} + d) = a\bar{Q} + b \tag{71}$$

Using (71) and standard properties of the Pauli matrices it is then easy to see that there are three kinds of fixed points or surfaces.

1. $\bar{Q}^{\pm} = \bar{x}_0^{\pm} I$, i.e. $\bar{x}_i = 0$. Here

$$\bar{x}_0^{\pm} = \frac{1}{2c} \left[ (a - d) \pm \sqrt{(a + d)^2 - 4} \right] \tag{72}$$

In deriving (72) we have used (70). These are two fixed points and requires

$$(a + d)^2 > 4 \tag{73}$$

The corresponding matrix

$$\begin{bmatrix} a & b \\ c & d \end{bmatrix} \tag{74}$$

is in the hyperbolic conjugacy class.

2. When $(a + d)^2 < 4$ there is a fixed two dimensional hypersurface defined by

$$\bar{x}_0 = \frac{a - d}{2c} \qquad \sum_{i=1}^{3} (\bar{x}_i)^2 = \frac{1}{4c^2} \left( 4 - (a + d)^2 \right) \tag{75}$$

and now the matrix in (74) is in the elliptic conjugacy class.

3. The case $(a + d)^2 = 4$ is marginal and corresponds to the parabolic conjugacy class.

It is now straightforward to see that the transformation (67) can be re-expressed in the following form for hyperbolic and elliptic transformations:

$$(Q' - \bar{Q}^+)(Q' - \bar{Q}^-)^{-1} = (c\bar{Q}^- + d)(Q - \bar{Q}^+)(Q - \bar{Q}^-)^{-1}(c\bar{Q}^+ + d)^{-1} \qquad (76)$$

For the elliptic conjugacy class, the $\bar{Q}^\pm$ in this equation refer to two antipodal points on the fixed 2-surface defined in (75).

To prove (76) we substitute (67) in the left hand side of this equation so that the left hand side becomes

$$\left[(a - c\bar{Q}^+)Q + (b - d\bar{Q}^+)\right]\left[(a - c\bar{Q}^-)Q + (b - d\bar{Q}^-)\right]^{-1} \qquad (77)$$

while the right hand side of (76) becomes

$$\left[(c\bar{Q}^- + d)Q - (c\bar{Q}^- + d)\bar{Q}^+\right]\left[(c\bar{Q}^+ + d)Q - (c\bar{Q}^+ + d)\bar{Q}^-\right]^{-1} \qquad (78)$$

For the hyperbolic case, $\bar{Q}^\pm$ are both proportional to the identity matrix. For the elliptic case, we we can take the antipodal points on the fixed 2-surface to be along the 3 axis without any loss of generality. For both these cases one can then explicitly check that

$$\bar{Q}^+ + \bar{Q}^- = \frac{a - d}{c} \cdot I \qquad \bar{Q}^+\bar{Q}^- = \bar{Q}^-\bar{Q}^+ = -\frac{b}{c} \cdot I \qquad (79)$$

Using these relations and the relation (70) it may be checked that (77) and (78) are equal term by term.

The relationship (67) can be iterated to yield

$$Q'^{(n)} = \left(I - (c\,Q_- + d)^n \alpha_\pm(Q)(c\,Q_+ + d)^{-n}\right)^{-1} \left(Q_+ - \left((c\,Q_- + d)^n \alpha_\pm(Q)(c\,Q_+ + d)^{-n}\right)Q_-\right), \qquad (80)$$

where

$$\alpha_\pm(Q) = \left[(Q - Q_+)(Q - Q_-)^{-1}\right].$$

which can be used to determine the trajectory of a point under successive transformations.

The Euclidean distance between two points represented by quaternions $Q_1$ and $Q_2$ are given by $\sqrt{|\text{Det}(Q_1 - Q_2)|}$. Equation (76) implies

$$\frac{\text{Det}(Q' - \bar{Q}^+)}{\text{Det}(Q' - \bar{Q}^-)} = R^2 \frac{\text{Det}(Q - \bar{Q}^+)}{\text{Det}(Q - \bar{Q}^-)} \qquad (81)$$

where

$$
\begin{aligned}
R^2 &= \frac{\text{Det}(cQ^- + d)}{\text{Det}(cQ^+ + d)} = \left(\frac{a + d - \sqrt{(a+d)^2 - 4}}{a + d + \sqrt{(a+d)^2 - 4}}\right)^2 \qquad &\text{(Hyperbolic)} \\
R^2 &= 1 \qquad &\text{(Elliptic)}
\end{aligned}
\qquad (82)
$$

The equation can be recursively applied to yield

$$\frac{\text{Det}(Q^{(n)} - \bar{Q}^+)}{\text{Det}(Q^{(n)} - \bar{Q}^+)} = R^{2n} \frac{\text{Det}(Q - \bar{Q}^+)}{\text{Det}(Q - \bar{Q}^+)} \qquad (83)$$

where $Q^n$ denotes the transformed point after $n$ such transformations. The expressions for $R^2$ then immediately implies that for hyperbolic transformations a point converges to one of the fixed points for large $n$, whereas for ellipic transformations, the point keeps moving on a circle of fixed radius around a point on the fixed surface.

Using the relation (80) we can calculate various physical quantities under a Floquet drive of this type, after $n$ cycles. The result depends on the conjugacy class of the transformation at the end of a cycle. However the general features will be similar to the $d = 1$ case, i.e. hyperbolic classes lead to a heating phase while elliptic classes lead to oscillatory phase [12, 14, 17].

When a cycle involves two different $SL(2, R)$ subgroups of the conformal group, e.g. two different $\Pi_\mu$'s, one can no longer represent the net transformation by a $SL(2, H)$ transformation with parameters proportional to the identity. The analysis of fixed points (surfaces) etc. needs to be re-done, and the result of such Floquet drives will be quite different.

Explicit results for physical quantities under Floquet dynamics will appear in a future publication.

## 5 Discussions

In this paper we initiated a program of studying periodic quantum dynamics of driven deformed conformal field theories in arbitrary number of dimensions, inspired by recent work in $1 + 1$ dimensions. More concretely, we studied the properties of driven $3 + 1$ dimensional CFTs using square protocols which involve evolution by non-commuting Hamiltonians in successive time intervals after a single drive cycle.

There are several points worth emphasizing regarding this approach. First, the method of exactly computing the dynamics works in arbitrary number of dimensions. We performed explicit calculations in $3 + 1$ dimensions in this work. But our use of quaternion formalism provides a simple way to perform these calculations all $d \leq 3$. Secondly, our method can be straightforwardly generalized to other drive protocols including continuous ones; we leave this as a subject of future work. For a class of drive protocols which uses same $\mu$ but different $\beta$ for each pulse within a drive cycle, we have shown how to obtain the necessary transformations for arbitrary number of cycles. This is similar to $d = 1$, and we can have dynamic control over the transition from the heating to the non-heating phase similar to that found in $d = 1$ driven CFTs [12, 14, 17]. Explicit results for multiple cycles of this type will appear in a future paper. For pulses with different $\mu$, the situation is less clear and this issue is left as a subject of future study.

While the power of conformal symmetry allows us to calculate observables and the details associated with the phase structure, perhaps it is useful to ponder over a simpler physical intuition of the underlying physics. In this regard, it is useful to consider *e.g.* free theories in (1+1)-dimensions. For free Bosons, the stress-tensor $T(w) = (-1/2) : \partial\phi(w)\partial\phi(w) :$ and for free Fermions it is given by $T(w) = (1/2) : \psi(w)\partial\psi(w) :$ and subsequently the Virasoro modes are extracted from the stress-tensor: $L_n = (1/(2\pi i)) \oint w^{n+1} dw T(w)$. The standard radial quantization in CFT chooses a Hamiltonian $H \sim L_0$, which corresponds to a Hamiltonian-density of the form: $w\partial\phi(w)\partial\phi(w)$ for Bosons and $w\psi(w)\partial\psi(w)$ for Fermions. The $w$-factor in front of the kinetic terms can be interpreted to be yielding a

red-shift physics for the system. Now, choosing a different quantization basically entails distinct choices for the Hamiltonian, involving $\{L_m, L_{-m}\}$-modes. Such choices will generate a non-trivial red-shift compared to the standard quantization and a thermal-phase may appear as a result of this.[4] Note, however, that this is too crude to pass as an argument since we have not factored in the inequivalent conjugacy classes and we have also been imprecise about where the "red-shift" matters, *i.e.* near the origin of the complex plane or near infinity. Nonetheless, it raises an an intriguing possibility of a more precise physical picture along these lines and we hope to address this in future.

For holographic CFT's, it would be insightful to get a dual gravitational description of the dynamics. This can be done from two different perspectives: One that is based on the explicit time-dependent Hamiltonian and the other that is based on the Floquet Hamiltonian. While the former yields a dynamical geometry, the latter is capable of describing static "patches" of the geometry which correspond to the different phases (heating and non-heating) and the phase boundary. For the $1 + 1$ dimensional case, the dynamical scenario has been explored in [7–9], while the Floquet-Hamiltonian based approach is discussed in [22]. In [7], a non-trivial dynamical horizon structure has been observed, whereas the different geometric patches of [22] were obtained by solving integral curves that are generated by the (appropriate combination of) bulk Killing vector that is dual to the CFT Floquet Hamiltonian. Interestingly, the integral curve approach is also explored in a completely different context of "bulk reconstruction" in which a local bulk observer in an AdS-geometry is described in terms of the CFT Hamiltonian, see *e.g.* [33,34]. In [35] the authors considered $SL(2, R)$ drives starting from thermal states in 2D holographic CFTs, which corresponds in the bulk to BTZ geometry with evolving horizons. The analog in higher dimensions using our set-up would correspond to driving AdS-Schwarzschild black holes. As a black hole is expected to be chaotic, it will also be interesting to study how semiclassical chaos as captured by out-of-time-ordered-correlators behave in higher dimensional driven CFTs. In the context of 2D CFTs these explorations have already been carried out in [36].

The higher dimensional generalization of the bulk geometric description is potentially very interesting. First, it is rather non-trivial to obtain horizons with structures in higher dimensional black holes. Secondly, it is also technically involved to obtain dynamical horizons, outside Vaidya-type geometries. Moreover, from the integral curve perspective, a local bulk observer seems readily describable in terms of the CFT Floquet Hamiltonian using the bulk Killing vectors that describe the $SL(2, C)$ sub-algebra of the conformal algebra, see for example appendix B for explicit expressions of the corresponding generators. We are currently exploring this aspect in detail.

Let us now offer some comments that are not necessarily limited to CFTs. It is evident that the crucial ingredients of our results are the following: (i) The existence of an $SL(2, C)$ algebra as a sub-algebra of the symmetry group of the system and (ii) a "quasi-primary" representation of the fields under this $SL(2, C)$.[5] Interestingly, scattering matrix elements in $(3 + 1)$ and $(2 + 1)$-dimensions can be recast into correlation functions of quasi-primary operators at null infinity known as the celestial sphere, see *e.g.* [38–40] for explicit details on this. These quasi-primaries form the continuous principal series, $\Delta = \frac{d-1}{2} + is$, in $(d+1)$-dimensions and $s$ is real-valued; we have collected some relevant and explicit formulae in appendix C for a more direct comparison. Even though the "CFT-spectrum" is continuous and complex-valued, the corresponding correlators are real-valued since it involves both

---

[4]Qualitatively, this is similar to the Rindler-physics, although it appears more subtle and layered.

[5]A similar statement holds for an $SL(2, R)$ algebra as well.

$\Delta$ and $\bar{\Delta}$. Thus, the presence of a heating and a non-heating phase along with a phase boundary is expected in this case as well. Note, however, that this entails a rather non-trivial quantization of the system: the Minkowski coordinates, $X^\mu$, are first mapped to a stereographic coordinates on the null sphere, $w$; subsequently, the $w$-plane is mapped to the cylinder by $\zeta = \exp(2\pi w/L)$. Identifying $\zeta = T + ix$ and analytically continuing $T \to iT_L$ yields the Lorentzian system (with Lorentzian time $T_L$) for which the phases are expected to exist for different conjugacy classes of the $\mathrm{SL}(2, C)$-transformations. It will be very interesting to explore this aspect as well as its connection with scattering matrix physics in detail, which we leave for future.

# 6   Acknowledgements

We would like to thank Bobby Ezhuthachan, Akavoor Manu, Masahiro Nozaki and Koushik Ray for discussions. The work of S.R.D. is supported by a National Science Foundation grant NSF-PHY/211673 and Jack and Linda Gill Chair Professorship. S.R.D. would like to thank Yukawa Institute of Theoretical Physics and Isaac Newton Institute for Mathematical Physics for hospitality during the completion of this work. KS thanks DST for support through SERB project JCB/2021/000030. AK is partially supported by CEFIPRA $6304 - 3$, DAE-BRNS 58/14/12/2021-BRNS and CRG/2021/004539 of Govt. of India. D.D. acknowledges support by the Max Planck Partner Group grant MAXPLA/PHY/2018577 and from MATRICS grant SERB/PHY/2020334.

# A   Transformation of the generators

In this section, we use infinitesimal coordinate transformations to obtain expressions of the generators in the deformed frame as eluded to in the main text. To this end, we recollect from the main text that the transformation of the generators $D$, $K_0$ and $P_0$ can be easily obtained as follows

$$
\begin{aligned}
U^{-1}\sigma_z U &= \tau_z = \cosh\theta\,\sigma_z + \sinh\theta\, i\sigma_x \\
U^{-1}\sigma_x U &= \tau_x = \cosh\theta\,\sigma_x - i\sigma_z \sinh\theta, \quad \tau_y = \sigma_y
\end{aligned} \tag{84}
$$

where $U = \exp[-i\theta(K_0 - P_0)/2]$ as defined in the main text and the last relation follows from the fact that $U$ commutes with $\sigma_y$. Eq. 84 therefore species the relations between the generators $D$, $K_t$ and $P_t$ and their deformed versions $D'$, $K_0'$ and $P_0'$. These are given by

$$
\begin{aligned}
D' &= D\cosh\theta - \frac{1}{2}(K_0 + P_0)\sinh\theta \\
K_0'[P_0'] &= \frac{1}{2}(1 + \cosh\theta)K_0[P_0] - \frac{1}{2}(1 - \cosh\theta)P_0[K_0] - D\sinh\theta
\end{aligned} \tag{85}
$$

where we have used Eq. 9 to obtain the generators from the Pauli matrices.

To construct rest of the transformed generators, we need to use their coordinate representation. For this, it is convenient to first consider an infinitesimal transformation $U_{\mathrm{inf}} = 1 - i\theta(K_0 - P_0)/2$. The transformation of coordinates $(\tau, x, y, z) \equiv (\tau, x_j)$ under

the action of $U_{\text{inf}}$ can be computed and yields

$$\tau' = U\tau = \tau - \frac{\theta}{2}(1 + r^2 - 2\tau^2), \quad x'_j = Ux_j = x_j(1 + \theta\tau) \tag{86}$$

The reverse transformation expressing $(\tau, x_j)$ in terms of $(\tau', x'_j)$ can be obtained from Eq. 86 and yields

$$\tau = \tau' + \frac{\theta}{2}(1 + (r')^2 - 2\tau'^2), \quad x_j = x'_j(1 - \theta\tau') \tag{87}$$

We now use the transformed coordinates to write the expressions of the generators. For example, one finds

$$P'_0 = -i\partial_{\tau'} = -i\partial_\tau + i\theta(\tau\partial_\tau + \sum_j x_j\partial_j) = P_0 - \theta D \tag{88}$$

where we have used Eq. 87. Note that this coincides with the infinitesimal version of Eq. 85 as expected; this can also be explicitly checked for $D$ and $K_0$.

Next we find the other generators where the use of the transformed coordinates provides us the infinitesimal version of the generators in the deformed frame. For example, one finds

$$P'_j = -i\partial_{x'_j} = -i(\partial_{x_j} + \theta(x_j\partial_\tau - \tau\partial_{x_j})) = P_j + \theta L_{0j} \tag{89}$$

$$K'_j = -i(x'_j(\tau'\partial_{\tau'} + \sum_j x'_j\partial_{x'_j}) - (r')^2\partial_{x'_j})$$

$$= -i(x_j(\tau\partial_\tau + \sum_j x_j\partial_{x_j}) - r^2\partial_{x_j}) - i\theta(\tau\partial_{x_j} - x_j\partial_\tau) = K_j - \theta L_{0j} \tag{90}$$

where $L_{\mu\nu} = i(x_\mu\partial_{x_\nu} - x_\nu\partial_{x_\mu})$ are the angular momentum generators. A simlar analysis yields

$$L'_{0j} = L_{0j} + \frac{\theta}{2}(P_j - K_j), \quad L'_{ij} = L_{ij}. \tag{91}$$

It can be easily checked that the generators obtained in Eqs. 89, 90, and 91, together with the infinitesimal version of $D$, $K_0$ and $P_0$ obtained from Eq. 85, satisfy the conformal algebra to $O(\theta)$.

Next, we look for the transformed generators for finite conformal transformation. Instead of a direct calculation via coordinate transformations which is cumbersome, we use the results obtained for infinitesimal transformations and the criteria that these generators must satisfy the conformal algebra. It turns out that these two criteria can uniquely specify the deformed generators. To see this, we first consider $P_j$. Eq. 89 dictates the infinitesimal version of $P'_j$; taking cue from this and Eq. 85, we write

$$P'_j = \frac{1}{2}(1 + \cosh\theta)P_j + \frac{1}{2}(1 - \cosh\theta)K_j + L_{0j}\sinh\theta \tag{92}$$

The first check to ensure that this is the correct form is to verify the commutator $[D', P_j] = iP'_j$. Written explicitly, using standard commutation relation of conformal generators, this yields

$$[D', P'_j] = i\Big(\frac{\cosh\theta(1 + \cosh\theta)}{2}P_j - \frac{\cosh\theta(1 - \cosh\theta)}{2}K_j - \frac{1}{2}\sinh^2\theta(P_j + K_j)$$

$$+ \sinh\theta L_{0j}\Big) = iP'_j \tag{93}$$

Next, we need to check the commutation relation between $P'_j$ and $K'_j$. For this one needs to know $K'_j$. Using Eq. 90 and taking cue from the form of $P'_j$, we write

$$K'_j = \frac{1}{2}(1 + \cosh\theta)K_j + \frac{1}{2}(1 - \cosh\theta)P_j - L_{0j}\sinh\theta \qquad (94)$$

This form ensures $[D', K'_j] = -iK'_j$ and this can be checked in a similar manner. The commutation of $K'_j$ and $P'_j$ can be computed to be

$$[K'_j, P'_j] = 2i\left(\frac{(1 + \cosh\theta)^2 - (1 - \cosh\theta)^2}{4}D - \sinh\theta(K_0 + P_0)\right) = 2iD' \qquad (95)$$

The commutation of $P'_0$ and $K'_j$ can also be checked and yields

$$\begin{aligned}
[K'_j, P'_0] &= 2i\left(\frac{(1 + \cosh\theta)^2 - (1 - \cosh\theta)^2}{4}L_{0j} + \frac{1}{2}\sinh\theta D(P_j - K_j)\right) \\
&= 2i\left(\cosh\theta\, L_{0j} + \sinh\theta(P_j - K_j)/2\right) = 2iL'_{0j} \qquad (96)
\end{aligned}$$

Note that the form of $L'_{0j} = \cosh\theta\, L_{0j} + \sinh\theta(P_j - K_j)/2$ so obtained is consistent with Eq. 91. Further it can be checked that $[D', L'_{0j}] = 0$. The other commutation relations involving $L'_{ij}$ and $L'_{0j}$ also holds provided we set $L'_{ij} = L_{ij}$. Thus the final expressions of the deformed generators, used in the main text, are given by

$$\begin{aligned}
D' &= D\cosh\theta - \frac{1}{2}(K_0 + P_0)\sinh\theta \\
K'_0 &= \frac{1}{2}(1 + \cosh\theta)K_0 - \frac{1}{2}(1 - \cosh\theta)P_0 - D\sinh\theta \\
P'_0 &= \frac{1}{2}(1 + \cosh\theta)P_0 - \frac{1}{2}(1 - \cosh\theta)K_0 - D\sinh\theta \\
K'_j &= \frac{1}{2}(1 + \cosh\theta)K_j + \frac{1}{2}(1 - \cosh\theta)P_j - L_{0j}\sinh\theta \\
P'_j &= \frac{1}{2}(1 + \cosh\theta)P_j + \frac{1}{2}(1 - \cosh\theta)K_j + L_{0j}\sinh\theta \\
L'_{0j} &= \cosh\theta\, L_{0j} + \frac{1}{2}\sinh\theta(P_j - K_j), \quad L'_{ij} = L_{ij} \qquad (97)
\end{aligned}$$

Thus Eq. 97 provides a complete description of the deformed CFT in terms of the new generators.

# B   Conformal Algebra in $d$-dimensions

In this appendix, for completeness, we briefly collect the well-known conformal algebra and the corresponding generators. We will closely follow the notations of [37]. The conformal algebra in $R^d$ is given by the $SO(d+1,1)$ algebra, with the following commutation relations:

$$[D, P_\mu] = iP_\mu\,, [D, K_\mu] = -iK_\mu\,, [K_\mu, P_\nu] = 2i\left(\delta_{\mu\nu}D - M_{\mu\nu}\right)\,, \qquad (98)$$

$$[M_{\mu\nu}, P_\alpha] = i(\delta_{\nu\alpha}P_\mu - \delta_{\mu\alpha}P_\nu)\,, [M_{\mu\nu}, K_\alpha] = i(\delta_{\nu\alpha}K_\mu - \delta_{\mu\alpha}K_\nu)\,, \qquad (99)$$

$$[M_{\alpha\beta}, M_{\mu\nu}] = i(\delta_{\alpha\mu}M_{\beta\nu} + \delta_{\beta\nu}M_{\alpha\mu} - \delta_{\beta\mu}M_{\alpha\nu} - \delta_{\alpha\nu}M_{\beta\mu})\,. \qquad (100)$$

The differential operator representation of this algebra is given by

$$P_\mu = -i\partial_\mu\,, D = -ix^\mu\partial_\mu\,, M_{\mu\nu} = -i(x_\mu\partial_\nu - x_\nu\partial_\mu)\,, \qquad (101)$$

$$K_\mu = -2ix_\mu x^\nu\partial_\nu + ix^2\partial_\mu\,. \qquad (102)$$

The corresponding Euclidean AdS spacetime can be described by the hyperboloid in $R^{d+1,1}$:

$$-(X^0)^2 + (X^1)^2 + \ldots (X^{d+1})^2 = -L^2 \; . \tag{103}$$

In the global patch, the AdS metric is given by

$$ds^2 = L^2 \left( \cosh^2 \rho \; d\tau^2 + d\rho^2 + \sinh^2 \rho \; dY_{d-1}^2 \right) \; , \quad Y \cdot Y = 1 \; . \tag{104}$$

The $SO(d+1,1)$ algebra is generated by the following Killing vectors in global coordinate:

$$D = -i\partial_\tau, \;\; M_{\mu\nu} = -i\left( Y_\mu \tfrac{\partial}{\partial Y^\nu} - Y_\nu \tfrac{\partial}{\partial Y^\mu} \right),$$

$$P_\mu = -ie^{-\tau}\left[ Y_\mu(\partial_\rho + \tanh\rho \; \partial_\tau) + \frac{1}{\tanh\rho}\nabla_\mu \right], \;\; \nabla_\mu = \tfrac{\partial}{\partial Y^\mu} - Y_\mu Y^\nu \tfrac{\partial}{\partial Y^\nu},$$

$$K_\mu = -ie^{\tau}\left[ Y_\mu(-\partial_\rho + \tanh\rho \; \partial_\tau) - \frac{1}{\tanh\rho}\nabla_\mu \right]. \tag{105}$$

One can check that on the boundary $\rho = \infty$ these Killing vectors generate via Eq.(2) the Lüscher-Mack term $K_\mu + P_\mu$.

# C  SL$(2,C)$, Quasi-primaries, Lorentz symmetry

In this appendix we collect some relevant formulae and basic statements regarding the SL$(2,C)$ action of the Lorentz group in $(3+1)$-dimensions. We will closely follow [38–41]. It is well-known that given a four-vector $X^\mu$, we can associate a hermitian matrix:

$$X = \begin{bmatrix} X^0 - X^3 & X^1 + iX^2 \\ X^1 - iX^2 & X^0 + X^3 \end{bmatrix} \; , \quad \det(X) = -X^\mu X_\mu \; . \tag{106}$$

Now, an SL$(2,C)$ matrix $\Theta$ acts on the hermitian matrix $X$ as:

$$X' = \Theta X \Theta^\dagger \; , \quad \Theta = \begin{bmatrix} a & b \\ c & d \end{bmatrix} \; , \quad \text{with} \quad ad - bc = 1 \; , \tag{107}$$

such that $\det(X') = \det(X).^6$

    A more detailed action can be obtained by noting that the Lorentz group in $(3+1)$-dimensions acts as the global conformal group on the celestial sphere at infinity. More explicitly, given the Minkowski coordinates $X^\mu$, $\mu = 0, \ldots 3$, the celestial sphere is defined by $\eta_{\mu\nu}X^\mu X^\mu = 0$, on which we can define: $w = (X^1 + iX^2)/(X^0 + iX^3)$. Under a Lorentz transformation: $X'^\mu = \Lambda^\mu_\nu X^\nu$, we get: $w' = (aw + b)/(cw + d)$, where $ad - bc = 1$ and $a, b, c, d$ are complex-valued. Subsequently, the $(3+1)$-dimensional scattering amplitudes can be expressed as celestial CFT correlators.

---

[6]In $(2+1)$-dimensions a similar description holds for a real-symmetric matrix [41]:

$$X = \begin{bmatrix} X^0 - X^2 & X^1 \\ X^1 & X^0 + X^2 \end{bmatrix} \; , \quad X' = \Lambda X \Lambda^T \; , \quad \Lambda \in SL(2,R) \; . \tag{108}$$

For an explicit realization, [38–40] has defined quasi-primary fields under this $SL(2,C)$ in the principal series, with a conformal dimension $\Delta = 1 + is$, where $s$ is real-valued.[7] Moreover, there is a precise state-operator correspondence as well. The states are described by $\left| h, \bar{h}, w, \bar{w} \right\rangle$ where $\{w, \bar{w}\}$ are the stereographic coordinates of the celestial sphere. Lorentz transformation, denoted by $U(\Lambda)$ acts on these states as follows:

$$U(\Lambda) \left| h, \bar{h}, w, \bar{w} \right\rangle = \frac{1}{(cw + d)^h} \frac{1}{(\bar{c}\bar{w} + \bar{d})^{\bar{h}}} \left| h, \bar{h}, \frac{aw + b}{cw + d}, \frac{\bar{a}\bar{w} + \bar{b}}{\bar{c}\bar{w} + \bar{d}} \right\rangle \ , \qquad (109)$$

$$\Lambda \in SL(2, C) \ , \quad h = \frac{1 + is - \sigma}{2} \ , \quad \bar{h} = \frac{1 + is + \sigma}{2} \ , \qquad (110)$$

where $\sigma$ denotes the helicity of the massless particle.

The corresponding basis quasi-primary states are given by

$$|s, \sigma, w = 0 = \bar{w}\rangle = \frac{1}{\sqrt{8\pi^4}} \int_0^\infty dE E^{is} |E, 0, 0, E, \sigma\rangle \ , \qquad (111)$$

such that the states are normalized as follows:

$$\langle s_1, \sigma_1, w_1, \bar{w}_1 | s_2, \sigma_2, w_2, \bar{w}_2 \rangle = \delta(s_1 - s_2)\, \delta^{(2)}(w_1 - w_2)\, \delta_{\sigma_1 \sigma_2} \ , \qquad (112)$$

where

$$|s, \sigma, w, \bar{w}\rangle = \left( \frac{1}{1 + w\bar{w}} \right)^{1+is} U(R(w, \bar{w})) |s, \sigma, 0, 0\rangle \ , \qquad (113)$$

in which $U(R(w, \bar{w}))$ are the unitary rotation operators. The action of the Lorentz group on these states then yields equation (109), which essentially is comprised of $SL(2, C)$ matrix multiplications. The corresponding operators, denoted by $\mathcal{O}_{h, \bar{h}}$ will evolve in the Heisenberg picture according to:

$$U(\Lambda)\mathcal{O}_{h, \bar{h}}(w, \bar{w}) U(\Lambda)^\dagger = \left( \frac{\partial \Lambda w}{\partial w} \right)^h \left( \frac{\partial \Lambda \bar{w}}{\partial \bar{w}} \right)^{\bar{h}} \mathcal{O}_{h, \bar{h}}(\Lambda w, \Lambda \bar{w}) \ . \qquad (114)$$

A very similar description exists in $(2 + 1)$-dimensions as well, we refer the interested Reader to *e.g.* [41] for more details on this.

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
