# Peer review of "Exactly Solvable Floquet Dynamics for Conformal Field Theories in Dimensions Greater than Two"

_SciPost Physics_

## Round 1 · Referee Report · Anonymous (Referee 1) · 2024-3-25

Strengths

Nice setup.

Weaknesses

Not well written. Many typos.

Report

This manuscript discusses the quantum dynamics of conformal field theory governed by SL2-deformed Hamiltonian in higher dimensions, which generalizes previous studies in 1+1D. The authors introduce quaternion to give an efficient method of calculating the time evolution in spatial dimensions not larger than three, which I found quite nice. Several quantities like Fidelity, two-point functions and one-point functions are calculated. Very similar to the result in 1+1D, there are two phases, heating and non-heating phases, which depends on whether the generator of the dynamics is in hyperbolic or elliptic class. This manuscript might fit Physics Core better than Physics.

Below are some comments and questions. I would like to hear the authors’ response first before making a final decision.

General comments:

1. The manuscript provides several concrete results on quench dynamics, while it does not contain a serious study on Floquet dynamics. Therefore I personally found the title of the manuscript misleading. I suggest that the author could at least change the title of section 4 to something like "quench dynamics", which is more informative for readers to know what it is talking about.

2. I understand that the formalism is applicable to both 2+1D and 3+1D CFT. I am curious why the authors directly consider 3+1D CFTs to do explicit calculations rather than 2+1D. Presumably, one can give a simpler and clearer geometric presentation of the results in 2+1D.

3. It seems that the manuscript is not written carefully. I often need to guess the meaning of notations. It also contains many typos, which I was only able to identify a subset of. Both affects the reading experience. It will be very helpful to readers if the authors could revise the manuscript more carefully.

Specific comments:

1. In eqn 1, the definition of Pi_mu involves X_mu. But there is no clear definition of X_mu.

2. The notation in Section 2.1 is hard to understand. For example, D’ and K0’, P0’ are not clearly defined. Also the notation K0[P0] is quite confusing. I suggest the authors to improve it a bit, e.g. clearly write down the definition/relation between un-primed and primed operators.

3. Eqn 21, setting beta = 0, we have Lambda_\pm = 0 and log \Lambda_0 to be reduced to -4w to have an equality. But I somehow wasn’t able to show log \Lambda_0 = -4w in this case. Is there a typo or I made a mistake by myself?

4. Maybe good to write \mu \neq \nu in Eqn 22 explicitly if this is a necessary assumption.

5. Eqn 23, the text and equations are mixed together.

6. Right now, the quench protocol (Eqn 34, 35, 36) is written in the section of fidelity, which is not easy for the reader to identify directly. I suggest that the authors can either use a few paragraph to introduce the setup at the beginning of section 4 before any specific discussions or somewhere else.

7. Eqn 33, typo in the denominator. Is it Delta instead of h?

8. Fig 5, how to understand that the energy peaks are moving in time? In 1+1D, one can have an intuitive understanding of the Floquet CFT results in terms of quasi-particle picture, as is discussed in Ref 31. Do we also have some simple explanation here?

9. Fig9, there seems to be a typo in the caption. Is it theta=0 or phi = 0?

10. In the discussion, naively, I thought the energy density does not have the omega prefactor. Maybe the author can explain it a bit more or clarify the meaning of the derivative?

  • validity: -
  • significance: -
  • originality: -
  • clarity: -
  • formatting: below threshold
  • grammar: -

Author:  Diptarka Das  on 2024-06-02  [id 4535]

(in reply to Report 1 on 2024-03-25)
Category:
remark

We thank the referee for their comments. We have updated our version on the arXiv (https://arxiv.org/abs/2311.13468) addressing all the suggestions and criticisms of the referee. However we have decided to withdraw the submission from SciPost.

---

## Round 1 · Referee Report · Anonymous (Referee 2) · 2024-4-2

Strengths

1-The method of exactly computing the dynamics works in arbitrary dimensions, with explicit calculations demonstrated in 3 + 1 dimensions, showcasing the versatility of the approach. The framework developed can be recursively applied to multiple cycles, allowing for the study of Floquet dynamics as a function of stroboscopic time, potentially revealing dynamical phases and transitions between them.

2-The draft provides insights into the behavior of driven conformal field theories in dimensions greater than two, particularly focusing on exactly solvable Floquet dynamics using square pulse protocols. It also explores different behaviors in the driven system based on the value of the deformation parameter $\beta$, such as exponential decays, oscillations, and power law decays, contributing to a comprehensive understanding of the system's dynamics.

3-The analysis of the energy density and one-point function amplitude in various regimes, including heating and non-heating phases, offers a detailed examination of the system's response to the drive protocol, highlighting features like localization and oscillations.

Weaknesses

1-This draft essentially uses the SU(1,1) subgroup structure of the high dimensional CFT and, therefore, does not contain enough novel results beyond
[6,12,14,15].

Report

This draft does not meet the acceptance criteria of Scipost Physics.

  • validity: ok
  • significance: low
  • originality: poor
  • clarity: high
  • formatting: good
  • grammar: good

Author:  Diptarka Das  on 2024-06-02  [id 4534]

(in reply to Report 2 on 2024-04-02)
Category:
remark

We thank the referee for their comments. We have updated our version on the arXiv (https://arxiv.org/abs/2311.13468 -- kindly consult this version for remarks below), however we have decided to withdraw the submission from SciPost.

For completeness we reply to the weakness pointed out by the referee : The novelty beyond a single $SU(1,1)$ beyond [6,12,14,15] is present when (i) deformation in different directions are considered, e.g., in case of the Fidelity computation (Section 4.1), general rule of composing Quaternions for different $SU(1,1)$s ( Section 4.5 and Appendix B) (ii) the structure of fixed points (in Euclidean signature) is different for SL(2,H) leading to richer dynamics.

---

## Editorial Decision

awaiting_resubmission